# Thymic dendritic cell-derived IL-27p28 promotes the establishment of functional bias against IFN-γ production in newly generated CD4+ T cells through STAT1-related epigenetic mechanisms

**Jie Zhang[1], Hui Tang[1], Haoming Wu[1], Xuewen Pang[1], Rong Jin[1]\*, Yu Zhang[1,2]\***

[1]Department of Immunology, School of Basic Medical Sciences, NHC Key Laboratory of Medical Immunology, Medicine Innovation Center for Fundamental Research on Major Immunology-related Diseases, Peking University, Beijing, China; [2]Institute of Life Sciences, Jinzhou Medical University, Jinzhou, China

## eLife Assessment

This study presents a **useful** reassessment of the potential role of dendritic cell-derived IL-27 p28 cytokine in the functional maturation of CD4+CD8- thymocytes, and CD4+ recent thymic emigrants. The evidence supporting the claims of the authors is **solid** and serves to reaffirm what has been previously described, with the overall advance in understanding the mechanism(s) responsible for the intrathymic functional programming of CD4+ T cells being limited.

**\*For correspondence:**
jinrong@bjmu.edu.cn (RJ);
zhangyu007@bjmu.edu.cn (YZ)

**Competing interest:** The authors declare that no competing interests exist.

**Abstract** The newly generated CD4 single-positive (SP) T lymphocytes are featured by enhanced IL-4 but repressed IFN-γ production. The mechanisms underlying this functional bias remain elusive. Previous studies have reported that CD4+ T cells from mice harboring dendritic cell (DC)-specific deletion of IL-27p28 display an increased capacity of IFN-γ production upon TCR stimulation. Here, we demonstrated that similarly altered functionality occurred in CD4SP thymocytes, recent thymic emigrants (RTEs), as well as naive T cells from either *Cd11c-p28*^f/f mice or mice deficient in the α subunit of IL-27 receptor. Therefore, DC-derived IL-27p28-triggered, IL-27Rα-mediated signal is critically involved in the establishment of functional bias against IFN-γ production during their development in the thymus. Epigenetic analyses indicated reduced DNA methylation of the *Ifng* locus and increased trimethylation of H3K4 at both *Ifng* and *Tbx21* loci in CD4SP thymocytes from *Cd11c-p28*^f/f mice. Transcriptome profiling demonstrated that *Il27p28* ablation resulted in the coordinated up-regulation of STAT1-activated genes. Concurrently, STAT1 was found to be constitutively activated. Moreover, we observed increased accumulation of STAT1 at the *Ifng* and *Tbx21* loci and a strong correlation between STAT1 binding and H3K4me3 modification of these loci. Of note, *Il27p28* deficiency exacerbated the autoimmune phenotype of *Aire*^-/- mice. Collectively, this study reveals a novel mechanism underlying the functional bias of newly generated CD4+ T cells and the potential relevance of such a bias in autoimmunity.

## Introduction

The thymus plays a crucial role in the phenotypic development and functional maturation of T cells. After positive and negative selection, T cell progenitors become CD4 or CD8 single positive (CD4SP or CD8SP) thymocytes that are self-tolerant. These SP thymocytes then emigrate to the periphery, known as recent thymic emigrants (RTEs), and 2–3 weeks later, RTEs develop into mature naive T cells (*Ashby and Hogquist, 2024*). Upon activation, CD4SP thymocytes and RTEs exhibit a reduced proliferation capacity and decreased production of interleukin-2 (IL-2), interferon-γ (IFN-γ), and tumor necrosis factor α (TNF-α) compared to mature naive T cells. However, they show increased secretion of IL-4, which is known as T helper (Th) 2 bias in immature T cells (*Hendricks and Fink, 2011*). This Th2 bias in CD4SP thymocytes and RTEs is thought to assist newly emigrated self-reactive T cells to respond appropriately to self-antigens (*Fink and Hendricks, 2011*). Furthermore, the low level of IFN-γ production in RTEs renders them more prone to becoming regulatory T cells, which also helps to prevent the induction of autoimmune diseases (*Bhaumik et al., 2013*). The low level of DNA methylation at the *Il4* locus and reduced expression of *Dnmts* contribute to the Th2 bias in CD4SP thymocytes and RTEs (*Berkley et al., 2013*). However, the mechanism behind repressed IFN-γ expression in favor of the Th2 bias in these immature T cells is elusive.

The heterodimeric cytokine IL-27 is composed of p28 (also named IL-27p28, or IL-30) and Epstein-Barr virus-induced gene 3 (EBI3), which signals through the IL-27R consisting of the IL-27Rα (also named WSX-1, TCCR) and the gp130 subunits. Elevated levels of IL-27 are associated with the pathogenesis of autoimmune diseases such as ankylosing spondylitis (AS; *Lin et al., 2015*), Behcet's disease (BD; *Shen et al., 2013*), and experimental arthritis (*Cao et al., 2008*). The IL-27-regulated differentiation and function of peripheral CD4⁺ T cells subsets Th1, Th17, and Treg have been proposed as an explanation for the pathogenesis of autoimmune disease (*Mei et al., 2021*), however, conflicting reports have been demonstrated in different settings. For example, it is consistent that IL-27 inhibits Th17 differentiation and function *in vitro* and *in vivo* in *Il27p28⁻/⁻* and *Ebi3⁻/⁻* experimental autoimmune encephalomyelitis (EAE) mice models (*Diveu et al., 2009*; *Kim et al., 2019*; *Liu et al., 2012*). Although IL-27 inhibits Treg differentiation *in vitro* (*Huber et al., 2008*; *Neufert et al., 2007*), and *Ebi3⁻/⁻* EAE models show increased numbers and suppressive functions of Tregs (*Liu et al., 2012*), Tregs in *Il27ra⁻/⁻* and Treg-specific *Il27ra⁻/⁻* mice loss the suppressive function (*Kim et al., 2019*; *Nguyen et al., 2019*; *Park et al., 2019b*). Furthermore, IL-27 was initially found to promote Th1 differentiation of naive CD4⁺ T cells under physiological conditions (*Pflanz et al., 2002*). However, naive CD4⁺ T cells derived from *Ebi3⁻/⁻* and *Il27p28⁻/⁻* mice exhibit increased production of IFN-γ. In addition, the absence of *Il27ra⁻/⁻* promotes pathological Th1 accumulation following *Trypanosoma cruzi* infection (*Hamano et al., 2003*). A subsequent study demonstrates that both the absence and overexpression of IL-27p28 lead to increased susceptibility to *T. gondii* infection, highlighting the role of IL-27p28 as an independent negative regulator of IFN-γ production in T cells (*Park et al., 2019a*; *Villarino et al., 2003*). Notably, deficiency in gp130 does not alter IFN-γ production in CD4⁺ T cells in a virus-infection model (*Harker et al., 2015*). These results indicate the intricate functionality of each subunit of IL-27 and IL-27R in peripheral CD4⁺ T cells, which complicates the interpretation of the autoimmune phenotype in deficient mice.

The structural analysis of IL-27 has demonstrated that IL-27p28, serving as a central subunit, possesses the ability to bind with EBI3, as well as the receptor subunits IL-27Rα (*Caveney et al., 2022*). Previous studies have reported IL-27p28 as an independent antagonist of gp130-mediated signaling (*Stumhofer et al., 2010*). In a ConA-induced liver damage model, CD4⁺ T cells from mice with DC-specific deletion of *IL27p28* (*Cd11c-p28^{f/f}* mice) exhibit a propensity for rapid and robust production of IFN-γ upon activation. This observation is attributed to the absence of IL-27p28 in the thymic environment of these mice (*Zhang et al., 2013*). These findings raise the intriguing possibility that thymic DC-derived IL-27p28 may contribute to shaping the Th1/Th2 balance of newly generated CD4SP thymocytes.

In this paper, we have demonstrated that DC-derived IL-27p28 plays a crucial role in maintaining the Th2 bias of CD4SP thymocytes. Our findings suggest that IL-27p28 regulates the DNA methylation status and histone modifications in the transcription regulatory regions of *Ifng* and *Tbx21* genes. These epigenetic modifications were further modulated by the low level of basal and phosphorylated signal transducer and activator of transcription 1 (STAT1) protein. Furthermore, we observed that the disruption of Th2 bias in CD4SP thymocytes due to IL-27p28 deficiency exacerbated autoimmune

diseases in *Aire^-/-* mice. Our results provide insights into the molecular mechanisms underlying the regulation of Th2 bias by DC-derived IL-27p28 and its potential role in autoimmune diseases.

## Results

### DC-specific deletion of IL-27p28 endows the newly generated CD4⁺ T cells with inherently enhanced capacity of IFN-γ production

Although IL-27 is widely reported to promote peripheral Th1 differentiation *in vitro*, CD4⁺ T cells from *Cd11c-p28^f/f* mice exhibit enhanced IFN-γ production, a capacity seemingly acquired in their development in the thymus (*Zhang et al., 2013*). To investigate the possibility that thymic DC-derived IL-27 shapes the functionality of newly generated CD4⁺ T cells, CD4SP thymocytes, CD4⁺ RTEs, and CD4⁺ naive T cells were purified from *Cd11c-p28^f/f* (cKO) mice and their wild-type (WT) littermates harboring a *Rag2*-GFP transgene, which allows easy tracking of newly generated thymocytes and RTEs (*Ashby and Hogquist, 2024*). The purified cells were then compared for cytokine expression in response to anti-CD3 and anti-CD28 stimulation under non-polarizing conditions. As expected, CD4⁺ naive T cells from cKO mice exhibited a significantly higher level of *Ifng* mRNA expression compared to WT cells (*Figure 1A*). Intracellular staining (*Figure 1B*) and ELISA assay of the culture supernatant (*Figure 1C*) confirmed enhanced IFN-γ production at the protein level. Notably, this difference was already detectable at the SP thymocyte stage, indicating that it was developmentally regulated. Another prominent feature of CD4SP thymocytes is the hyperproduction of Th2 cytokines (*Fink and Hendricks, 2011*; *Makar et al., 2003*). Despite the profound impact of IL-27p28 deficiency on IFN-γ production, we observed no significant difference in IL-4 expression between cKO and WT T cells (*Figure 1A and C*). Moreover, there was no significant difference in the expression of IL-2 and TNF-α between cKO and WT T cells (*Figure 1A*, *Figure 1—figure supplement 1*). These results suggest that DC-specific deletion of IL-27p28 specifically enhances IFN-γ production in newly generated T cells without affecting Th2 cytokines.

Next, we investigated the expression of T-box transcription factor (T-bet) and GATA3, which are the master regulators of Th1 and Th2 differentiation, respectively (*Fang et al., 2022*). The differential impact of IL-27p28 deficiency on IFN-γ and IL-4 production was accompanied by an increase in Tbx21 expression but no significant alteration in Gata3 expression (*Figure 1D*). Western blotting and flow cytometry confirmed the upregulation of T-bet protein expression (*Figure 1E and F*). Furthermore, we extended our analyses onto CD4⁺ T cells at steady state or under polarizing conditions. Higher basal levels of *Ifng* and *Tbx21* transcripts were detected in CD4SP thymocytes and naive T cells from cKO mice than the counterparts from WT mice even without stimulation (*Figure 1G*), indicating the constitutive activation of these loci. When cultured under polarizing conditions, these cells were effectively induced to differentiate into Th1, Th2, Th17 and Treg lineages. In contrast to the previous report of different differentiation potentials of RTEs versus naive T cells (*Hendricks and Fink, 2011*), CD4SP thymocytes, RTEs and naive T cells showed similar capacity to differentiate into effector cells. Moreover, cKO and WT cells were equally positioned for the differentiation (*Figure 1—figure supplement 2A–F*), suggesting that IL-27p28 deficiency does not affect effector cell differentiation under optimal conditions. Moreover, the frequency of IFN-γ-, and granzyme B-producing CD8⁺ T cells under Th0 conditions was comparable between cKO and WT T cells (*Figure 1—figure supplement 2G*). Taken together, these data support that IL-27p28 is actively involved in the functional maturation of developing CD4SP thymocytes, biasing them away from the Th1 lineage.

### *Il27ra*-deficient CD4⁺ T cells are similarly predisposed for potent IFN-γ production

Apart from acting as a subunit of IL-27, IL-27p28 have been reported to possess activities independent of IL-27, by acting on its own or coupling with other proteins, such as IL-Y (p28/p40, binding to IL-27Rα/IL-12Rβ1 to activate STAT3; *Flores et al., 2015*) and CLY (p28/CLF, binding to gp130/IL-6Rα to activate STAT1 and STAT3; *Tormo et al., 2013*). To ascertain whether the phenotype in the *Cd11c-p28^f/f* mice results from defective IL-27Rα signaling, we investigated the functional characters of T lineage cells in mice deficient in *Il27ra*. Reminiscent to what was observed in *Il27p28*-deficient mice, increased expression of *Ifng* and *Tbx21* were detected in *Il27ra^-/-* CD4SP thymocytes and naive CD4⁺ T cells, whereas *Il4*, *Il2*, and *Gata3* transcripts were equally present in WT and *Il27ra^-/-* cells (*Figure 2A*).

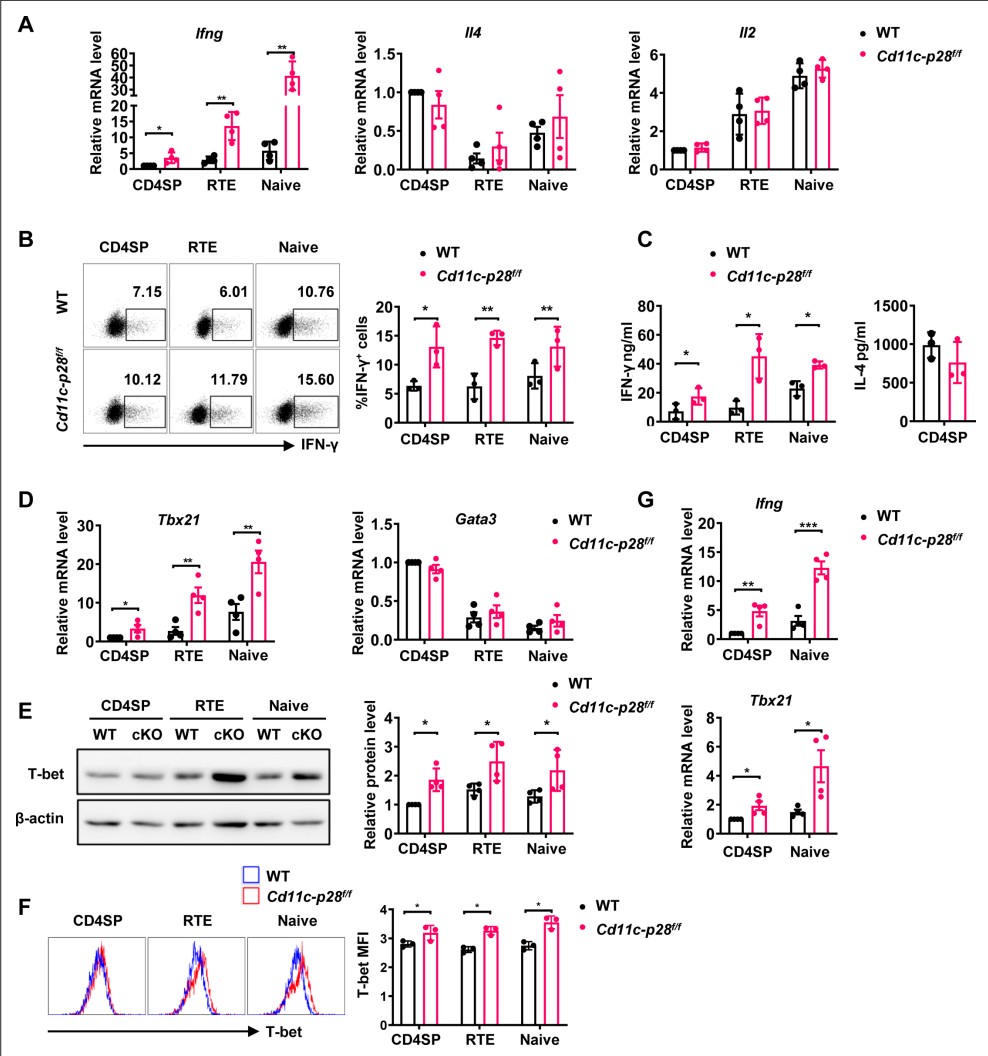

**Figure 1.** Elevated IFN-γ production and T-bet expression in *Cd11c-p28*[f/f] mice initiated from CD4SP thymocytes stage. (**A**) CD4SP (GFP⁺CD4⁺CD8⁻CD44^lo) thymocytes, CD4⁺ RTEs (GFP⁺CD4⁺CD8⁻CD25⁻CD44^lo) and CD4⁺ naive (GFP⁻CD4⁺CD8⁻CD25⁻CD44^lo) T cells were sorted from 6- to 8-week-old *Cd11c-p28*[f/f] mice and WT littermates, stimulated with plate coated anti-CD3 (2 µg/mL) and soluble anti-CD28 (1 µg/mL) for 12 hr. mRNA levels of *Ifng*, *Il4*, and *Il2* were determined by qPCR. Data: mean ± SD (n=4, duplicates). (**B**) Sorted cells were cultured under Th0 conditions for 3 days. The frequency of IFN-γ-producing CD4⁺ T cells were measured by intracellular staining. Representative dot plots (left) and statistical data (right, mean ± SD, n=3). (**C**) Supernatants from 3-day cultures were analyzed for IFN-γ and IL-4 by ELISA. Data: mean ± SD (n=3). (**D**) mRNA levels of *Tbx21* and *Gata3* in sorted cells were determined by qPCR. Data: mean ± SD (n=4, duplicates). (**E–F**) T-bet protein levels were assessed by western blot (**E**) and flow cytometry (**F**) after 3-day culture. Data: mean ± SD (n=3). (**G**) Freshly sorted cells were lysed in Trizol, and *Ifng* and *Tbx21* mRNA levels were determined by qPCR. Data: mean ± SD (n=4, duplicates). Statistical differences: * *p*<0.05, ** *p*<0.01, *** *p*<0.001 (Student's *t*-test).

The online version of this article includes the following source data and figure supplement(s) for figure 1:

**Source data 1.** PDF file containing original western blots for *Figure 1E*, indicating the relevant bands and treatments.

**Source data 2.** Original files for western blot analysis displayed in *Figure 1E*.

**Figure supplement 1.** The production of IL-2 and TNF-α was not altered during CD4SP thymocytes maturation for p28 deficiency.

**Figure supplement 2.** *In vitro* differentiation of CD4⁺ T cells under polarized conditions is unaffected by p28 deficiency.

Intracellular staining confirmed the enhanced IFN-γ production by *Il27ra*-deficient cells stimulated under non-polarizing conditions (*Figure 2B*). IFN-γ production by *Il27ra*[-/-] and WT cells, however, was comparable under Th1 polarizing conditions (*Figure 1—figure supplement 2H*). The similar phenotype shared by *Il27p28* and *Il27ra* knockout mice indicates that the altered T cell functionality in the absence of IL-27p28 is indeed a reflection of disrupted IL-27Rα signaling.

## IL-27p28 deficiency induces permissive epigenetic changes at *Ifng* and *Tbx21* loci

Developmentally regulated transcriptional activation and repression of genes or cell-type specific expression patterns are largely achieved by modifying the chromatin template at a gene locus (*Ivashkiv, 2018*). Murine neonatal CD4[+] T cells have been documented to be poised to rapid Th2 cytokine secretion due to the pre-existing hypomethylation at a key Th2 cytokine regulatory region (*Rose et al., 2007*). In other reports, Th2 polarization in adult SP thymocytes and RTEs is attributable to active recruitment of DNA methyltransferases and increased H3K4 methylation at *Il4* locus (*Berkley et al., 2013*; *Makar et al., 2003*). In addition, DNA methylation and histone modification are found to be important for the control of *Ifng* and *Tbx21* expression (*Fang et al., 2022*; *Friedman et al., 2023*; *Jones and Chen, 2006*). Therefore, we next evaluated the contribution of epigenetic modifications to the enhanced IFN-γ production in the absence of IL-27p28. Firstly, we performed bisulfite genomic sequencing on DNA of CD4 SP thymocytes from *Cd11c-cre p28*[f/f] mice and WT littermates. We focused on the nine CpG sites (*Figure 3A*) most proximal to the *Ifng* transcription start site, among which the –53 CpG has previously been suggested to play a key role in IFN-γ repression in Th2 effector cells (*Jones and Chen, 2006*). DNA methylation was significantly reduced at three CpG sites (–53,–34 and +16 site) of the *Ifng* locus in CD4SP thymocytes from *Cd11c-cre p28*[f/f] mice (*Figure 3B*). On the contrary, the five CpG sites in the *Il4* promoter region (*Figure 3C and D*), whose demethylation was associated with high level IL-4 production (*Lee et al., 2002*), showed no difference between cKO and WT CD4SP thymocytes.

As much as histone modifications are concerned, histone H3 lysine 27 trimethylation (H3K27me3) and H3 lysine 9 trimethylation (H3K9me3) are typically associated with repressed *Ifng* and *Tbx21* expression in Th cells, whereas histone H3 lysine 4 trimethylation (H3K4me3) enhances gene expression (*Fang et al., 2022*; *Friedman et al., 2023*). To evaluate the histone trimethylation modifications at *Ifng*, *Tbx21*, *Il4*, and *Gata3* loci in CD4SP thymocytes from WT and *Cd11c-p28*[f/f] mice, chromatin immunoprecipitation was performed using antibodies against H3K4me3, H3K9me3 and H3K27me3. H3K4me3 was found to be accumulated at *Ifng* and *Tbx21* loci in cKO CD4SP thymocytes, while its level in *Gata3* and *Il4* loci was comparable to that of wild type cells (*Figure 3E*). On the other hand, neither the occupancy of H3K9me3 and H3K27me3 at *Ifng*, *Tbx21*, *Il4*, and *Gata3* loci was altered (*Figure 3F and G*), nor were the total protein levels of H3K4me3, H3K9me3 H3K27me3 or the mRNA level of enzymes catalyzing the formation of these modifications (*Figure 3—figure supplement 1A–C*). These results suggest a link between up-regulated *Ifng* and *Tbx21* expression and altered epigenetic modifications in the absence of IL-27p28.

## Coordinated up-regulation of STAT1-activated genes in CD4SP thymocytes from *Cd11c-p28*[f/f] mice

To better understand the molecular mechanisms underlying the altered functionality, the transcriptional profile of CD4SP thymocytes from *Cd11c-p28*[f/f] mice was explored by RNA-Seq and compared with that of the WT counterparts. The two sets of differentially expressed genes (DEGs) were subjected to analyze for gene ontology (GO) biological process terms and Kyoto Encyclopedia of Genes and Genomes (KEGG) pathways. The up-regulated DEGs showed significant enrichment in genes related to host defense to viral infection, including interferon response and Jak-STAT signaling (*Figure 4A*). No meaningful hit, on the other hand, was revealed for the down-regulated gene set (data not shown).

Given that STAT1 is a key mediator downstream of IL-27 signaling (*Philips et al., 2022*), we were particularly interested in the altered expression of STAT1-regulated genes in the absence of IL-27p28. A previous study by *Hirahara et al., 2015* has compared the transcriptomes of wild-type and STAT1-deficient CD4[+] T cells stimulated by IL-27. Using their data (GSE65621), we compiled a list of genes potentially under regulation of STAT1. The DEGs were then examined for overlaps with this gene set. Surprisingly, over 50% of up-regulated DEGs (69 out of 137) fell into the category of STAT1-activated

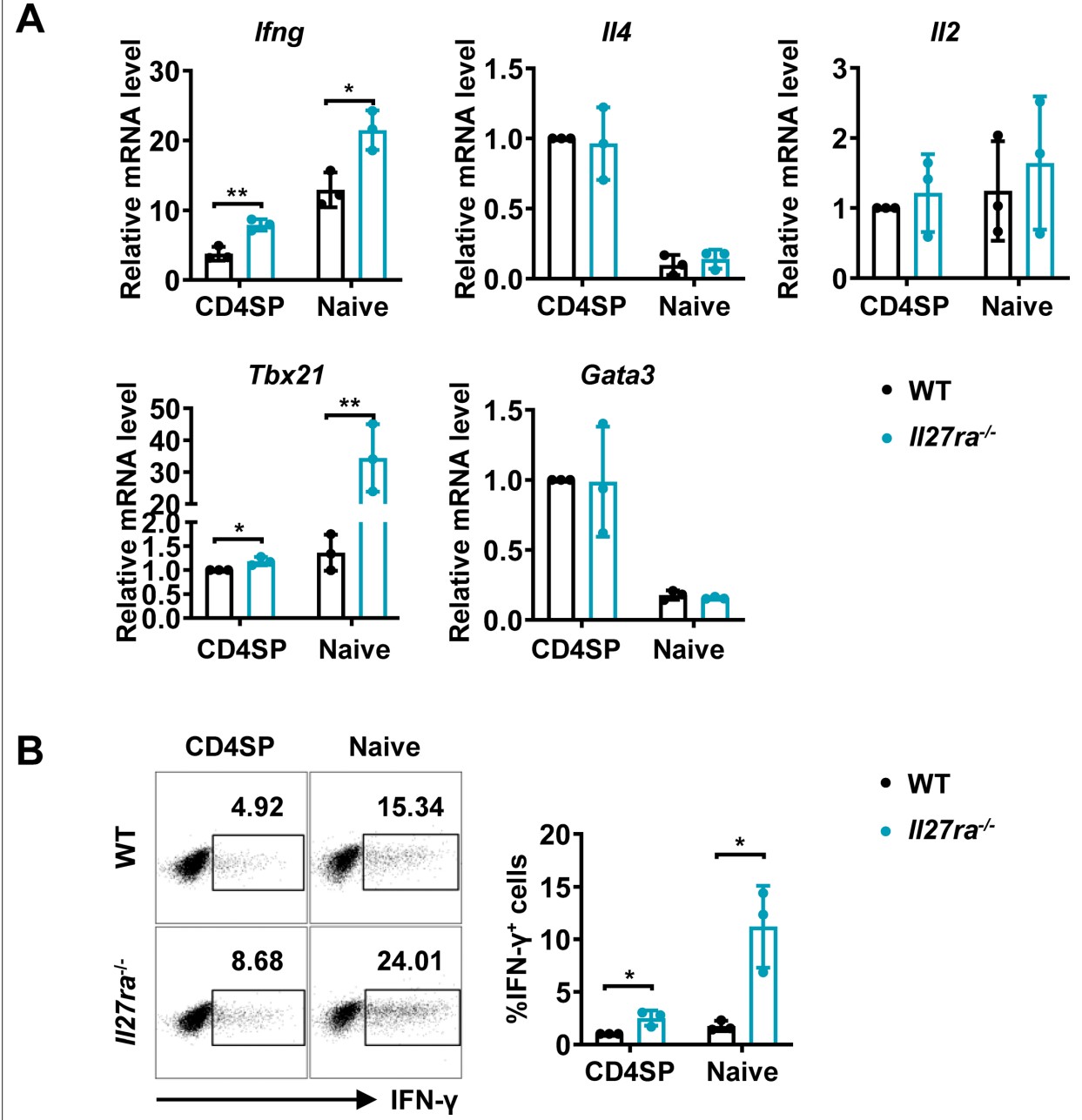

**Figure 2.** Enhanced IFN-γ production and T-bet expression in *Il27ra*[-/-] mice initiated at the CD4SP thymocytes stage. (**A**) CD4SP thymocytes and naive CD4[+] T cells were isolated from *Il27ra*[-/-] and WT mice, stimulated with plate-coated anti-CD3 (2 µg/mL) and soluble anti-CD28 (1 µg/mL) for 12 hours. mRNA levels of *Ifng*, *Il4*, *Il2*, *Tbx21*, and *Gata3* were determined by qPCR. Data: mean ± SD (n=3, duplicates). (**B**) CD4SP thymocytes and CD4[+] naive T cells were cultured under Th0 conditions for 3 days. The frequency of IFN-γ-producing CD4[+] T cells were analyzed by intracellular staining. Representative dot plots (left) and quantification (right, mean ± SD, n=3). Significance: * $p<0.05$, ** $p<0.01$ (Student's *t*-test).

genes, whereas none of them belonged to STAT1-suppressed genes (*Figure 4B*). To validate the RNA-Seq data, qPCR was carried out for a group of overlapping genes, including *Gm12250*, *Oasl2*, *Oas3*, *Parp14*, *Ifit3*, *Usp18*, *Igtp*, *Irf1*, *Ifi44*, *Oas2*, *Rsad2*, *Il12r*. As shown in *Figure 4C*, all these genes showed differential expression between WT and cKO CD4SP thymocytes. To further illustrate modest but coordinate changes in the expression of STAT1-regulated genes, gene set enrichment analysis (GSEA) was performed without imposing an arbitrary fold-change cutoff. Indeed, significant enrichment of STAT1-activated genes was detected in the transcriptome of CD4SP thymocytes from the

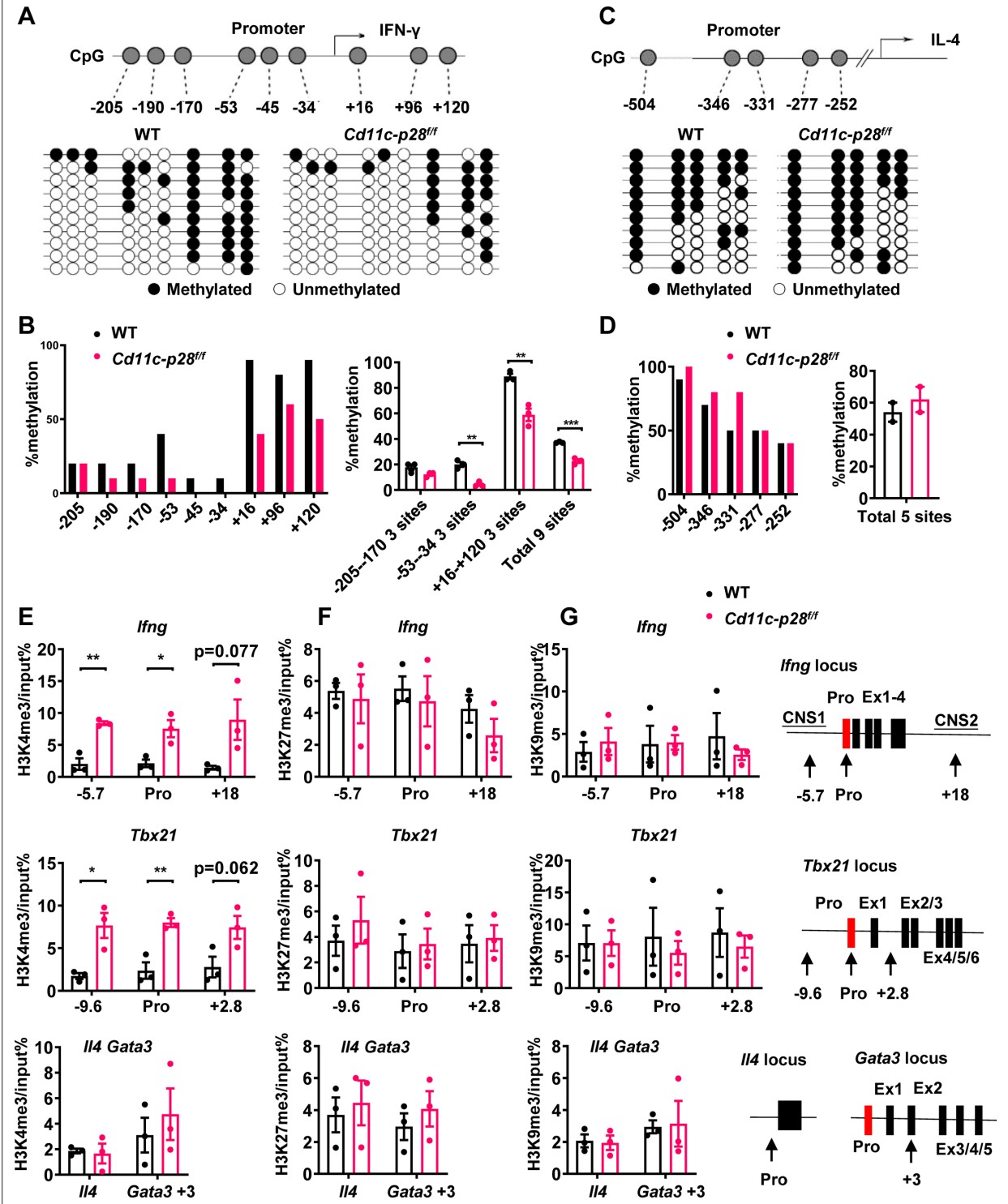

**Figure 3.** Distinct DNA and H3K4 methylation patterns at *Ifng* and *Tbx21* promoter regions in CD4SP thymocytes from *IL27p28*-deficient mice. (**A**) DNA methylation analysis of nine CpG sites in the *Ifng* promoter using sodium bisulfite-treated genomic DNA from GFP⁺CD4⁺CD8⁻CD44ˡᵒ CD4SP thymocytes. Each row represents a sequenced allele (n=10 clones from one of the three independent experiments). Filled (●) and open (O) circles denote methylated and unmethylated cytosine, respectively. (**B**) Left: Percent methylation at individual CpG sites from one representative experiment. Right: Average methylation of three adjacent site groups (group1: −205,−190, −170; group2: −53,−45, −34; group3:+16,+96,+120) and all CpG sites (mean ± SD, n=3). (**C**) DNA methylation analysis of five CpG sites upstream of the *Il4* transcription start site using sodium bisulfite-treated genomic DNA from GFP⁺CD4⁺CD8⁻CD44ˡᵒ CD4SP thymocytes. Each row represents a sequenced allele (n=10 clones from the two independent experiments). Filled (●)

*Figure 3 continued on next page*

*Figure 3 continued*

and open (O) circles denote methylated and unmethylated cytosine, respectively. (**D**) Left: graphs show the percentage of methylation at each individual site (left panel) or all CpG sites (right panel). (**E–G**) Histone trimethylation analysis in freshly isolated CD4SP thymocytes from *IL27p28*-deficient and WT mice. ChIP-qPCR was performed using antibodies against H3K4me3 (**E**), H3K27me3 (**F**), and H3K9me3 (**G**). qPCR primers targeted promoter and trans-regulatory regions of *Ifng*, *Tbx21*, *Il4*, and *Gata3*. Data: mean ± SEM (n=3, duplicates). Significance: * $p<0.05$; ** $p<0.01$ (Student's $t$-test). Abbreviation: pro., promoter.

The online version of this article includes the following source data and figure supplement(s) for figure 3:

**Figure supplement 1.** Basal levels of H3K4me3, H3K27me3, H3K9me3, and methylation-related enzymes are unaffected by p28 deficiency.

**Figure supplement 1—source data 1.** PDF file containing original western blots for *Figure 3—figure supplement 1*, indicating the relevant bands and treatments.

**Figure supplement 1—source data 2.** Original files for western blot analysis displayed in *Figure 3—figure supplement 1*.

knockout mice (NES = 1.67, NOM *p*-val=10⁻¹⁶, *Figure 4D*). Therefore, IL-27p28 deficiency resulted in an enhanced STAT1 activity.

To reveal functionally important connections of the DEGs, protein-protein interaction network was generated using Network Analyst (*Xia et al., 2015*). After removal of isolated and loosely connected nodes, the rest of the proteins encoded by the DEGs were imported into Network Analyst for network construction. As shown in *Figure 4E*, STAT1 was centrally positioned in the network with the highest degree of interactions. Taken together, these results indicate that IL-27p28 deficiency is strongly associated with an altered transcriptional program featured by enhanced expression of STAT1-activated genes.

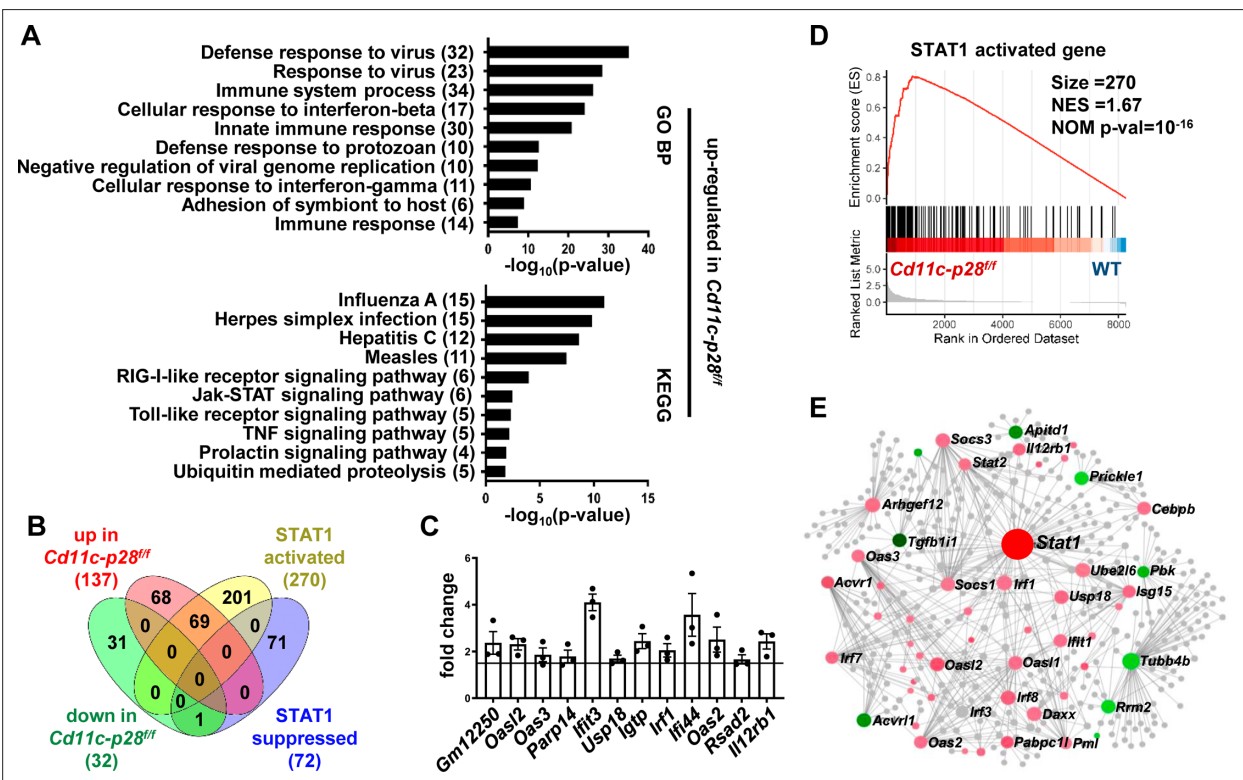

**Figure 4.** Increased expression of STAT1-activated genes in CD4SP thymocytes from *Cd11c-p28^f/f* mice. RNA-seq was performed to analyze the transcriptome of CD4SP thymocytes from *Cd11c-p28^f/f* and WT mice. (**A**) Top 10 enriched Gene Ontology (GO) biological process and Kyoto Encyclopedia of Genes and Genomes (KEGG) pathways for up-regulated differentially expressed genes (DEGs) in *Cd11c-p28^f/f* mice. (**B**) Overlap between DEGs in *Cd11c-p28^f/f* mice and STAT1-activated and suppressed genes. Numbers indicate overlapping genes in each category. (**C**) Validation of RNA-Seq results by qPCR for representative up-regulated genes. Data: fold change in *Cd11c-p28^f/f* versus WT mice (mean ± SEM, n=3, duplicates). (**D**) Gene Set Enrichment Analysis (GSEA) showing coordinated upregulation of STAT1-activated genes in *Cd11c-p28^f/f* CD4SP thymocytes. (**E**) Protein-protein interaction network of DEGs. Nodes represent proteins; edges indicate interactions. Larger nodes denote higher interaction degrees. Red: up-regulated; green: down-regulated; gray: non-DEGs connected to the network (added by Network Analyst).

## Constitutive activation of STAT1 in CD4SP thymocytes in the absence of IL-27p28

The distinct transcriptional signature prompted us to examine the activation status of STAT1 in freshly isolated CD4SP thymocytes from *Cd11c-p28*<sup>f/f</sup> mice and the WT littermates. Total thymocytes were stained for CD4 and CD8, followed by intracellular staining with antibodies against phosphorylated STATs. Significant levels of STAT1 Y701 phosphorylation were observed in cKO but not in WT cells. On the other hand, both cKO and WT cells were stained negative for phosphorylated STAT3 and STAT4 (*Figure 5A*). Phosphorylated and total STAT proteins were also analyzed by western blotting in purified CD4SP thymocytes, RTEs and naive CD4$^+$ T cells. While the total protein of STAT1 was comparable between cKO and WT cells in all populations examined, elevated levels of phosphorylated STAT1 Y701 were constantly detected in cKO cells. Moreover, the phosphorylation of STAT1 Y701 appeared to be developmentally regulated, increasing progressively from CD4SP thymocytes to naive T cells (*Figure 5B*). Another predominant phosphorylation site in STAT1 is serine 727, which is crucial for maximal STAT1 transcription activity (*Barnholt et al., 2009*). Basal levels of S727 phosphorylation were also observed in CD4SP thymocytes, but with no difference in WT and cKO cells (data not shown). As much as STAT3 and STAT4 were concerned, neither total nor phosphorylated STAT3 and STAT4 were found to be altered with IL-27p28 ablation (*Figure 5B*), indicating that the impact was specific for STAT1.

STAT1 plays an important role in the development of Th1 response by regulating *Tbx21* and *Ifng* expression (*Fang et al., 2022*). To explore the correlation of the increased STAT1 activation with the enhanced T-bet expression and IFN-γ production, we assessed STAT1 binding on the *Tbx21* and *Ifng* loci by ChIP assay. Possibly due to the relatively low abundance of activated STAT1, poor consistency was observed from one experiment to another with freshly isolated CD4SP thymocytes. Therefore, we chose to treat the cells with anti-CD3 and anti-CD28 for 3 days prior to the assay. Under such conditions, STAT1 was found to be markedly accumulated in the promoter and regulatory regions of both *Tbx21* and *Ifng* loci of cKO mice (*Figure 5C*). Notably, there were strong positive correlations between the levels of STAT1 binding and H3K4me3 (Pearson's correlation coefficient for *Ifng* locus $r=0.917$, $p=0.010$ and *Tbx21* locus $r=0.991$, $p<0.001$, *Figure 5D*), indicating the implication of STAT1 in the epigenetic modifications of the loci. Together, these data suggest that IL-27p28-medaited signal normally imposes an inhibitory effect on STAT1 activation, which in turn helps the establishment of functional bias against the Th1 lineage.

## Augmented autoimmune phenotype of *Aire*-deficient mice in the absence of IL-27p28

The physiological significance remains elusive of the bias against the Th1 effector functions for CD4SP thymocytes and RTEs. One hypothesis proposes that it favors the development of peripheral tolerance by avoiding a more inflammatory and potential detrimental response to self-antigens (*Cunningham et al., 2018*; *Fink and Hendricks, 2011*; *Hendricks and Fink, 2011*). The disruption of such a bias in *Cd11c-p28*<sup>f/f</sup> mice provided a good model to test this hypothesis. Flow cytometric analysis of the peripheral T cell compartment demonstrated a significant increase of CD44<sup>hi</sup>CD62L<sup>-</sup> activated T cells with a concomitant decrease of CD44<sup>lo</sup>CD62L<sup>+</sup> naive T cells in the cKO mice compared to the WT littermates (*Figure 6A*). Nevertheless, the cKO mice displayed no obvious signs of autoimmunity up to the age of 24–30 weeks when assessed by auto-antibodies against double-strand DNA (*Figure 6B*) and tissue pathology (*Figure 6C*). Occasionally, small lymphoid foci were observed in the lung of cKO mice. But the CD4$^+$ T cells recovered from the lung tissues displayed a similar cytokine secretion profile to that of WT mice except for the intrinsic augmentation of IFN-γ production (*Figure 6D*). Therefore, IL-27p28 deficiency alone is insufficient to drive the development of autoimmunity.

We next sought to determine whether IL-27p28 deficiency would affect the autoimmune phenotype in the *Aire*<sup>-/-</sup> mouse, which are predisposed to developing autoimmunity due to a defect in clonal deletion of autoreactive T cells (*van Laar et al., 2022*; *Wang et al., 2021a*). To this end, *Cd11c-p28*<sup>f/f</sup> mice were crossed with *Aire*<sup>-/-</sup> mice to generate double knockout mice. Histological examination revealed increased lymphocyte infiltration and more severe structural distortion in the lung and stomach in double knockout mice in comparison to mice deficient for Aire only (*Figure 6C*). In addition, the double knockout mice exhibited an elevated level of autoantibodies to double-strand DNA, although the difference did not reach a statistical significance ($p=0.068$) possibly due to the limited

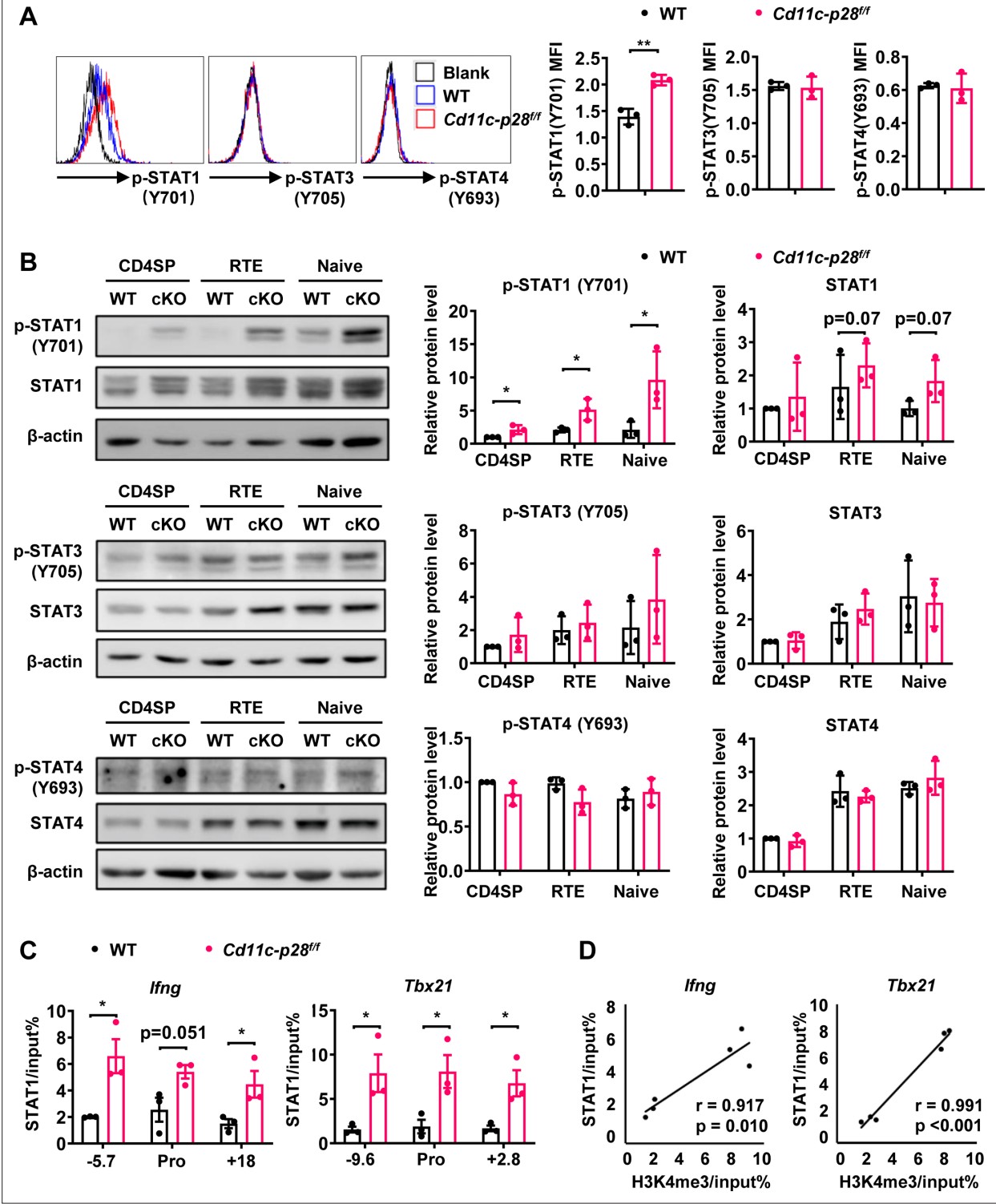

**Figure 5.** Enhanced STAT1 activation in CD4SP thymocytes from *Cd11c-p28^f/f* mice. (**A**) Intracellular staining of freshly isolated thymocytes from *Cd11c-p28^f/f* and WT mice using antibodies against phosphorylated STAT1 (Y701), STAT3 (Y705), and STAT4 (Y693). Representative histograms for CD4SP thymocytes (left) and mean fluorescence intensity (MFI) from three independent experiments (right, mean ± SD). (**B**) Western blot analysis of total and phosphorylated STAT1 (Y701), STAT3 (Y705), and STAT4 (Y693) in purified CD4SP thymocytes, CD4⁺ RTEs, and naive CD4⁺ T cells from *Cd11c-p28^f/f* and WT mice. Representative blots (left) and relative protein levels quantified by densitometry and normalization to β-actin (right, mean ± SD, n=3). (**C**) Increased STAT1 binding to promoter and regulatory regions of *Tbx21* and *Ifng* loci in *Cd11c-p28^f/f* mice. (**D**) Correlation between STAT1 binding and H3K4me3 levels at *Tbx21* and *Ifng* loci. Significance: * *p*<0.05; ** *p*<0.01.

*Figure 5 continued on next page*

*Figure 5 continued*

The online version of this article includes the following source data and figure supplement(s) for figure 5:

**Source data 1.** PDF file containing original western blots for *Figure 5B*, indicating the relevant bands and treatments.

**Source data 2.** Original files for western blot analysis displayed in *Figure 5B*.

**Figure supplement 1.** SOCS3 levels in CD4$^+$ T cells from *p28*-deficient mice.

**Figure supplement 1—source data 1.** PDF file containing original western blots for *Figure 5—figure supplement 1*, indicating the relevant bands and treatments.

**Figure supplement 1—source data 2.** Original files for western blot analysis displayed in *Figure 5—figure supplement 1*.

**Figure supplement 2.** The Stat1-dependent hyper-transcription of *Ifng* and *Tbx21* in p28 deficient CD4$^+$ T cells is not rescued by IFN-γ blockade.

**Figure supplement 2—source data 1.** PDF file containing original western blots for *Figure 5—figure supplement 2*, indicating the relevant bands and treatments.

**Figure supplement 2—source data 2.** Original files for western blot analysis displayed in *Figure 5—figure supplement 2*.

sample size (*Figure 6B*). When CD4$^+$ T cells recovered from the lung tissues were analyzed for IFN-γ, IL-4, and IL-17 production, the double knockout mice were virtually indistinguishable from the *Aire$^{-/-}$* mice, except for increased IFN-γ production (*Figure 6D*).

Immunological anergy constitutes an important mechanism for peripheral tolerance. Two studies have identified a subset of naturally occurring Foxp3$^-$CD44$^{hi}$CD73$^{hi}$FR4$^{hi}$ polyclonal CD4$^+$ T cells in healthy hosts. They are enriched for self-antigen-specific T cell antigen receptors and represent functionally anergic cells (*ElTanbouly and Noelle, 2021*; *Kalekar et al., 2016*). We wondered whether the Th1 bias induced by IL-27p28 deficiency would affect the generation of the anergic population. As shown in *Figure 6E*, the CD4$^+$Foxp3$^-$CD44$^{hi}$CD73$^{hi}$FR4$^{hi}$ anergic cells were significantly reduced in the double KO mice, which was accompanied by an increase in the CD4$^+$Foxp3$^-$CD44$^{hi}$CD73$^{lo}$FR4$^{lo}$ effector/memory cells. Taken together, these data support that the bias against the Th1 lineage in newly generated T cells provides a unique mechanism to render autoreactive cells exquisitely sensitive to tolerance induction.

## Discussion

Th2 polarization in adult SP thymocytes and RTEs is believed to be maintained through epigenetic modification at the *Il4* locus. However, the regulation of the opposing cytokine IFN-γ in this context remains unclear. We reported here that thymic DC-derived IL-27p28 played a key role in the establishment of the repressive status of the *Ifng* locus in newly generated CD4$^+$ T cells. As such, the loss of IL-27p28 in DCs endowed these cells with increased capacity of IFN-γ production upon TCR engagement. Transcriptome profiling demonstrated coordinated up-regulation of STAT1-activated genes in the absence of IL-27p28. Indeed, enhanced phosphorylation of STAT1 was observed in CD4SP thymocytes from *Cd11c-p28$^{f/f}$* mice, which was accompanied by the accumulation of STAT1 at the promoter and enhancer regions of *Ifng* and *Tbx21*. Epigenetic analyses indicated reduced DNA methylation at the *Ifng* locus and increased trimethylation of H3K4 at both *Ifng* and *Tbx21* loci in the absence of IL-27p28. Moreover, the H3K4me3 modification was shown to be strongly correlated with STAT1 binding to these loci. These data support a hypothesis that the exposure of developing thymocytes to IL-27p28 induces repressive epigenetic changes at the *Ifng* and *Tbx21* loci, possibly through antagonizing STAT1 activation, ultimately leading to an attenuated IFN-γ response of newly generated CD4SP thymocytes (*Figure 6F*).

The present study revealed a significantly elevated level of phosphorylated STAT1 and up-regulation of a variety of STAT1-regulated genes in CD4SP thymocytes from *Cd11c-p28$^{f/f}$* mice. These findings collectively suggest that IL-27p28 plays an important role in the fine-tuning of STAT1 signaling pathways during thymocytes development (*Twohig et al., 2019*). As IL-27 is known to activate multiple STAT proteins, including STAT1, STAT3, STAT4, and STAT5 following engagement of the gp130/IL-27Rα receptor (*Hirahara et al., 2015*; *Philips et al., 2022*), the hyperactivation of STAT1 in the absence of IL-27p28 is somehow unexpected. We examined the potential impact of IL-27 deficiency on the expression of suppressors of cytokine signaling 3 (SOCS3), which has been shown to have a regulatory role in IL-27 signaling (*Owaki et al., 2006*). Comparable levels of SOCS3 were detected in CD4SP thymocytes and naive CD4$^+$ T cells from *Cd11c-p28$^{f/f}$* mice and their littermates

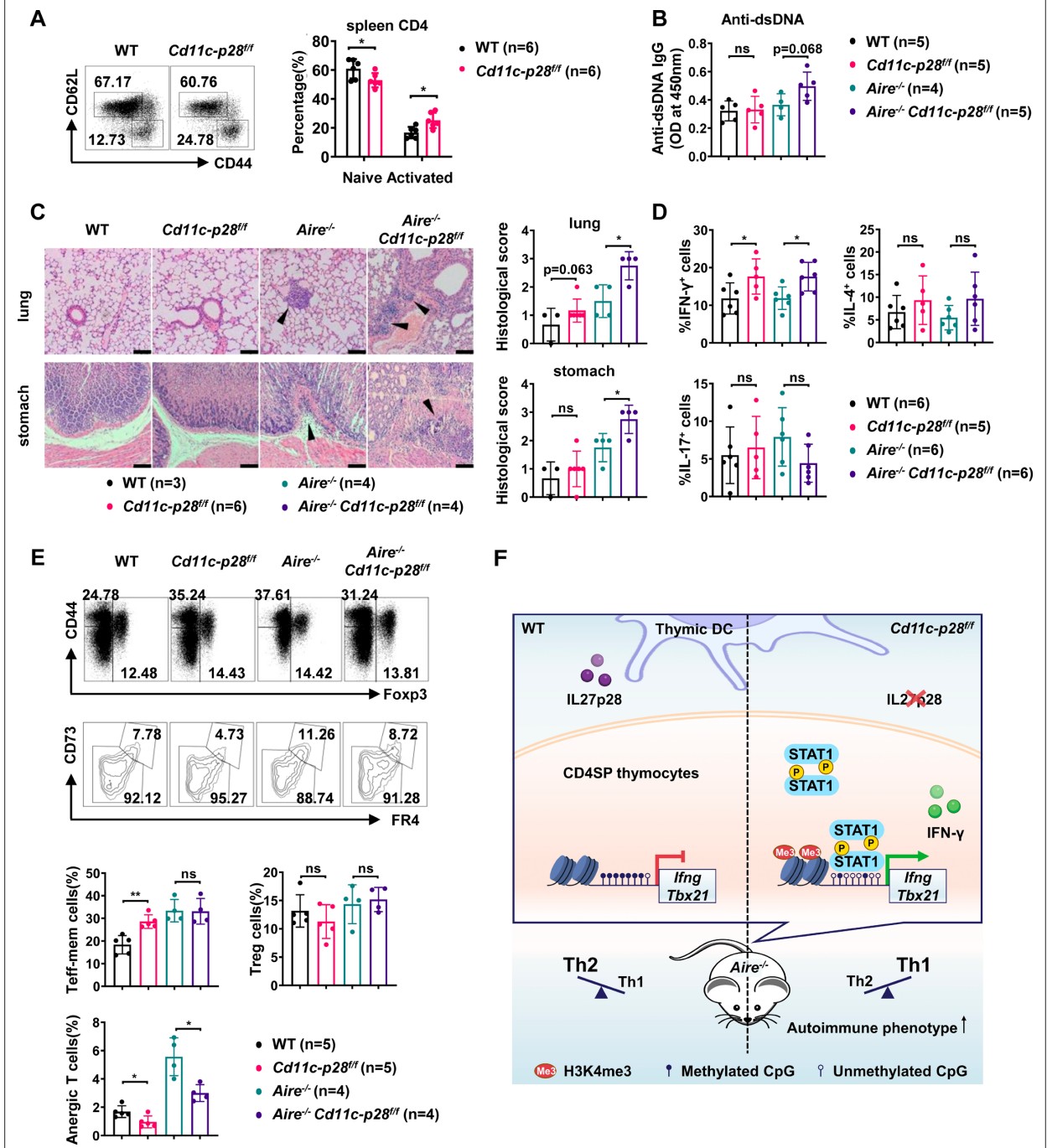

**Figure 6.** Exacerbated autoimmune responses in *Aire-/-* mice in the absence of IL27p28. (**A**) CD44 and CD62L expression in CD4+ T cells from splenocytes of 6–8 week-old WT (n=6) and *Cd11c-p28f/f* (n=6) mice. Representative dot plots (left) and percentages of CD44loCD62L+ naive and CD44hiCD62L- activated T cells are shown (right, mean ± SD). (**B–D**) WT, *Cd11c-p28f/f*, *Aire-/-*, and *Cd11c-p28f/fAire-/-* mice (24–30 weeks old) were analyzed. Each symbol represents one mouse (mean ± SD). (**B**) Serum anti-dsDNA antibody levels (ELISA). (**C**) H&E staining (left) and histological scores (right) of the lung and stomach. Arrows mark lymphocytic infiltrates. Scale bar = 100 μm. (**D**) Percentages of IFN-γ+, IL-4+, and IL-17A+ CD4+ T cells in lung tissue after PMA/ionomycin stimulation. (**E**) Splenocytes from 12-week-old mice were stained for CD44, Foxp3, CD73, and FR4. Percentages of anergic (CD4+Foxp3-CD44hiCD73hiFR4hi), effector/memory (CD4+Foxp3-CD44hiCD73loFR4lo), and regulatory (CD4+Foxp3+) T cells are shown. (F) Schematic summary of the study. * *p*<0.05; ** *p*<0.01; ns, not significant.

(*Figure 5—figure supplement 1*). We also explored the possibility that the constitutive activation of STAT1 might be the secondary effect of enhanced IFN-γ production (*Iwata et al., 2017*; *Singhania et al., 2019*). Addition of IFN-γ antibodies to the culture of CD4SP thymocytes and naive CD4⁺ T cells, however, showed no significant effect on phosphorylated STAT1 levels in knockout cells (*Figure 5—figure supplement 2A–B*). Thus, STAT1 hyperactivation is unlikely due to loss of SOCS3 expression or positive feedback of IFN-γ signaling. Of note, it has been previously demonstrated that IL-27p28 on its own can act as a natural antagonist of gp130-mediated signaling triggered by IL-6, IL-11, and IL-27 (*Stumhofer et al., 2010*). Consistent with this finding, *Chong et al., 2014* reported that IL-27p28 was able to inhibit IL-27-induced Th1 differentiation by reducing the phosphorylation of STAT1 and STAT3. Therefore, we speculate that IL-27p28 in the thymus may primarily functions to suppress STAT1 activation induced by other cytokines.

A recent study demonstrated that both recombinant murine and human IL-27 induced the expression of antiviral proteins *Oas1*, *Oas2*, *Oas3*, *Oasl1*, and *Oasl2* in epidermal keratinocytes. IL-27 signaling leads to OAS2 expression in a manner dependent on IL27Rα and STAT1, but independent of STAT2. Apart from their antiviral activity, OAS proteins are also involved in cell growth, differentiation, and apoptosis (*Castelli et al., 1997*; *Huang et al., 2022*; *Salzberg et al., 1997*). The genomic mutation and methylation of most OAS genes have been shown to alter their expression levels, which is associated with the levels of infiltrated CD4⁺ T cells and CD8⁺ T cells in the tumor microenvironment (*Gao et al., 2022*). Secreted OAS2 has also been observed to inhibit CD3 zeta chain expression in T cells (*Dar et al., 2016*). In our current study, we observed an increase in the transcription of *Oasl2*, *Oas2*, and *Oas3* in CD4SP thymocytes from *Cd11c-p28^f/f* mice, indicating that IL-27p28 signaling also involved in the fine-tuning of these OAS protein expressions. Whether these molecules play a role in thymocyte development remains to be clarified.

The biased functionality of newly generated CD4⁺ T cells has been proposed to be important for the induction of peripheral tolerance to self-antigens (*Cunningham et al., 2018*; *Fink and Hendricks, 2011*; *Hendricks and Fink, 2011*). Despite the much enhanced capacity of IFN-γ production by CD4⁺ T cells and a significant increase of CD44^hiCD62L⁻ activated T cells, *Cd11c-p28^f/f* mice displayed no apparent signs of autoimmunity under steady-state conditions. This is consistent with the result of our previous study in which negative selection was found to proceed normally in *Cd11c-p28^f/f* mice using the RIP-mOVA and OT-II models (*Tang et al., 2016*). We further interrogated the impact of the disrupted functional bias in *Aire^-/-* mice. Aire induces ectopic expression of self-antigens in medullary thymic epithelial cells to mediate negative selection of autoreactive thymocytes. Its mutation causes a rare life-threatening autoimmune disease affecting multiple organs (*van Laar et al., 2022*; *Wang et al., 2021a*). In comparison to *Aire^-/-* mice, *Cd11c-p28^f/f Aire^-/-* mice showed elevated levels of double-strand DNA autoantibodies, increased lymphocytes infiltration, and reduced anergic cells. Therefore, although disruption of the functional bias in newly generated T cells is not sufficient to drive autoimmunity, it accelerates disease development in hosts poised for autoimmunity.

In further support of a direct link between IL-27 and Aire, a recent study revealed that IL-27 production was inhibited in Aire-overexpressing murine dendritic cells (*Zou et al., 2021*). The two subunits IL-27p28 and EBI3 of human IL-27 are usually coordinately expressed in antigen-presenting cells, while mouse IL-27p28 can be secreted independently to inhibit the activities of IL-27 signaling (*Pflanz et al., 2002*; *Stumhofer et al., 2010*). The distinct production profile of IL-27p28 between humans and mice and our finding of the critical role of IL-27p28 in the maintenance of functional bias in newly generated CD4⁺ T cells may explain the much weaker autoimmune phenotype in *Aire^-/-* mice than *AIRE*-mutated human subjects. In addition, recent studies have reported elevation of serum IL-27p28 (also known as IL-30) in patients with prostate cancer (*Sorrentino et al., 2019*), psoriasis (*Omar et al., 2021*), and obesity (*Wang et al., 2021b*). It would be interesting to explore the involvement of altered functionality of newly generated CD4⁺ T cells under these pathological conditions.

Taken together, our data demonstrate that DC-derived IL-27p28 serves as an endogenous inhibitor for STAT1 hyper-activation in developing thymocytes. By regulating DNA and H3K4 methylation levels at the promoter and transcription regulatory regions of *Ifng* and *Tbx21*, it is critically involved in the establishment of the functional bias against IFN-γ production by newly generated CD4⁺ T cells. Disruption of this mechanism exacerbates autoimmune phenotypes in hosts predisposed for autoimmunity.

# Materials and methods

## Key resources table

| Reagent type (species) or resource | Designation | Source or reference | Identifiers | Additional information |
|---|---|---|---|---|
| Strain, strain background (*Mus musculus*) | *Cd11c-p28*^f/f mice(C57BL/6) | provided by Dr. Zhinan Yin from Jinan University (Guangzhou, China) | | |
| Strain, strain background (*Mus musculus*) | *Il27ra*^-/- mice (C57BL/6) | provided by Dr. Zhinan Yin from Jinan University (Guangzhou, China) | | |
| Strain, strain background (*Mus musculus*) | *Rag2*p-EGFP mice (C57BL/6) | This paper | | FVB-Tg (*Rag2*-EGFP) 1Mnz/J mice were purchased from Jackson Laboratory (Bar Harbor, ME) and were backcrossed for 10 generations onto the C57BL/6 background |
| Strain, strain background (*Mus musculus*) | *Aire*^-/- mice (C57BL/6) | provided by Yangxin Fu (University of Chicago, IL) | | |
| Antibody | PE-Cy7-conjugated anti-mouse CD4 (RM4-5) | BD Biosciences | Cat#: 561099 RRID:AB_394461 | FACS (5 µL per test) |
| Antibody | PE- conjugated anti-mouse CD8a (53–6.7) | BD Biosciences | Cat#: 561095 RRID:AB_394571 | FACS (5 µL per test) |
| Antibody | APC-conjugated anti-mouse CD8a (53–6.7) | BD Biosciences | Cat#: 561093 RRID:AB_398527 | FACS (5 µL per test) |
| Antibody | APC-conjugated anti-mouse IL-2 (JES6-5H4) | BD Biosciences | Cat#: 562041 RRID:AB_398555 | FACS (5 µL per test) |
| Antibody | PE-Cy7-conjugated anti-mouse TNF-α (MP6-XT22) | BD Biosciences | Cat#: 561041 RRID:AB_396761 | FACS (5 µL per test) |
| Antibody | PE-conjugated anti-mouse Stat1 (pY701) (4 a) | BD Biosciences | Cat#: 612564 RRID:AB_399855 | FACS (20 µL per test) |
| Antibody | PerCP-Cy5.5-conjugated anti-mouse Stat3 (pY705) (4/P-STAT3) | BD Biosciences | Cat#: 560114 RRID:AB_1645335 | FACS (20 µL per test) |
| Antibody | Alexa Fluor 488-conjugated anti-mouse Stat4 (pY693) (38/p-Stat4) | BD Biosciences | Cat#: 558136 RRID:AB_397051 | FACS (20 µL per test) |
| Antibody | PE- conjugated anti-mouse CD25 (PC61.5) | eBioscience | Cat#: 12-0251-82 RRID:AB_465607 | FACS (5 µL per test) |
| Antibody | APC-conjugated anti-mouse CD25 (PC61.5) | eBioscience | Cat#: 17-0251-82 RRID:AB_469366 | FACS (5 µL per test) |
| Antibody | PE-conjugated anti-mouse CD44 (IM7) | eBioscience | Cat#: 12-0441-82 RRID:AB_465664 | FACS (5 µL per test) |
| Antibody | APC-conjugated anti-mouse CD44 (IM7) | eBioscience | Cat#: 17-0441-82 RRID:AB_469390 | FACS (5 µL per test) |
| Antibody | FITC-conjugated anti-mouse FR4 (eBio12A5) | eBioscience | Cat#: 11-5445-82 RRID:AB_842799 | FACS (5 µL per test) |
| Antibody | PerCP-eFluor710-conjugated anti-mouse CD73 (eBioTY/11.8) | eBioscience | Cat#: 46-0731-82 RRID:AB_10853356 | FACS (5 µL per test) |
| Antibody | FITC-conjugated anti-mouse IL-4 (BVD6-24G2) | eBioscience | Cat#: 11-7042-82 RRID:AB_465388 | FACS (5 µL per test) |
| Antibody | PE-conjugated anti-mouse IL-17A (eBio17B7) | eBioscience | Cat#: 12-7177-81 RRID:AB_763582 | FACS (5 µL per test) |
| Antibody | APC-conjugated anti-mouse FOXP3 (3G3) | eBioscience | Cat#: MA5-16224 RRID:AB_2537742 | FACS (5 µL per test) |
| Antibody | biotin-conjugated anti-mouse CD8a (53–6.7) | eBioscience | Cat#: 13-0081-82 RRID:AB_466346 | FACS (5 µL per test) |
| Antibody | PE-conjugated anti-mouse IFN-γ (XMG1.2) | BioLegend | Cat#: 505807 RRID:AB_315401 | FACS (5 µL per test) |

*Continued on next page*

*Continued*

| Reagent type (species) or resource | Designation | Source or reference | Identifiers | Additional information |
|---|---|---|---|---|
| Antibody | PerCP-Cy5.5-conjugated anti-mouse T-bet (4B10) | BioLegend | Cat#: 644805 RRID:AB_1595488 | FACS (5 µL per test) |
| Antibody | Anti-phospho-STAT1(Tyr701) (Rabbit polyclonal) | Cell Signaling Technology | Cat#: 9167 RRID:AB_561284 | WB (1:1000) |
| Antibody | anti-phospho-STAT-1(S727) (Rabbit polyclonal) | Cell Signaling Technology | Cat#: 8826 RRID:AB_2773718 | WB (1:1000) |
| Antibody | anti-STAT1 (Rabbit polyclonal) | Cell Signaling Technology | Cat#: 14994 RRID:AB_2716759 | WB (1:1000) ChIP (1:50) |
| Antibody | anti-phospho-STAT3 (Tyr705) (Rabbit polyclonal) | Cell Signaling Technology | Cat#: 9131 RRID:AB_331588 | WB (1:1000) |
| Antibody | anti-STAT3 (Rabbit polyclonal) | Cell Signaling Technology | Cat#: 4904 RRID:AB_331269 | WB (1:2000) |
| Antibody | anti-phospho-STAT4 (Tyr693) (Rabbit polyclonal) | Cell Signaling Technology | Cat#: 4143 RRID:AB_10545742 | WB (1:1000) |
| Antibody | anti-STAT4 (Rabbit polyclonal) | Cell Signaling Technology | Cat#: 2653 RRID:AB_2255156 | WB (1:1000) |
| Antibody | anti-SOCS3 (Rabbit polyclonal) | Cell Signaling Technology | Cat#: 52113 RRID:AB_2799408 | WB (1:1000) |
| Antibody | anti-β-Actin (Rabbit polyclonal) | Cell Signaling Technology | Cat#: 4970 RRID:AB_2223172 | WB (1:1000) |
| Antibody | H3K4me3 (Rabbit polyclonal) | Millipore | Cat#: 07–473 RRID:AB_1977252 | WB (1:5000) ChIP (2 µL/test) |
| Antibody | H3K27me3 (Rabbit polyclonal) | Millipore | Cat#: 07–449 RRID:AB_310624 | WB (1:5000) ChIP (4 µL/test) |
| Antibody | H3K9me3 (Rabbit polyclonal) | Abcam | Cat#: ab8898 RRID:AB_306473 | WB (1:1000) ChIP (2–4 µg/test) |
| Antibody | Functional grade monoclonal antibodies for murine CD3 (145–2 C11) | eBioscience | Cat#: 16-0031-82 RRID:AB_468847 | Cell culture (2 µg/mL) |
| Antibody | Functional grade monoclonal antibodies for murine CD28 (37.51) | eBioscience | Cat#: 16-0281-82 RRID:AB_468921 | Cell culture (1 µg/mL) |
| Antibody | neutralizing anti-IL4 (11B11) | eBioscience | Cat#: 16-7041-81 RRID:AB_469208 | Cell culture (10 µg/mL) |
| Antibody | neutralizing anti-IFN-γ (clone XMG1.2) | eBioscience | Cat#: 16-7311-81 RRID:AB_469242 | Cell culture (10 µg/mL) |
| Sequence-based reagent | *Ifng*-F | This paper | PCR primers | TCAAGTGGCATAGATGTGGAAGAA |
| Sequence-based reagent | *Ifng*-R | This paper | PCR primers | TGGCTCTGCAGGATTTTCATG |
| Sequence-based reagent | *Il4*-F | This paper | PCR primers | ACAGGAGAAGGGACGCCAT |
| Sequence-based reagent | *Il4*-R | This paper | PCR primers | GAAGCCCTACAGACGAGCTCA |
| Sequence-based reagent | *Ifng* gene promoter CpG sites semi-nested PCR1-F | This paper | PCR primers | GGTGTGAAGTAAAAGTGTTTTTAGAGAATTTTAT |
| Sequence-based reagent | *Ifng* gene promoter CpG sites semi-nested PCR1-R | This paper | PCR primers | CAATAACAACCAAAAACAACCATAAAAAAAAACT |
| Sequence-based reagent | *Ifng* gene promoter CpG sites semi-nested PCR2-F | This paper | PCR primers | GGTGTGAAGTAAAAGTGTTTTTAGAGAATTTTAT |
| Sequence-based reagent | *Ifng* gene promoter CpG sites semi-nested PCR2-R | This paper | PCR primers | CCATAAAAAAAAACTACAAAACCAAAATACAATA |

*Continued on next page*

*Continued*

| Reagent type (species) or resource | Designation | Source or reference | Identifiers | Additional information |
|---|---|---|---|---|
| Sequence-based reagent | *Il4* gene promoter CpG sites semi-nested PCR-F | This paper | PCR primers | GGATCCACACGGTGCAAAGAGAGACCC |
| Sequence-based reagent | *Il4* gene promoter CpG sites semi-nested PCR-R | This paper | PCR primers | TCGGCCTTTCAGACTAATCTTATCAGC |
| Peptide, recombinant protein | Recombinant murine IL-2 | R&D Systems | Cat. #: 402 ML | Cell culture (2 ng/mL) |
| Peptide, recombinant protein | Recombinant murine IL-4 | R&D Systems | Cat. #: 404 ML | Cell culture (20 ng/mL) |
| Peptide, recombinant protein | Recombinant murine IL-6 | R&D Systems | Cat. #: 406 ML | Cell culture (10 ng/mL) |
| Peptide, recombinant protein | Recombinant murine IL-12 | R&D Systems | Cat. #: 419 ML | Cell culture (10 ng/mL) |
| Peptide, recombinant protein | recombinant human TGF-β1 | R&D Systems | Cat. #: 7754-BH | Cell culture (1 ng/mL) |
| Commercial assay or kit | IFN-γ ELISA kits | BioLegend | Cat. #:430807 | |
| Commercial assay or kit | IL-4 ELISA kits | BioLegend | Cat. #: 431107 | |
| Commercial assay or kit | EZ DNA methylation kit | Zymo Research | Cat. #: D5005 | |
| Commercial assay or kit | Pierce Agarose Chip Kit | Thermo Fisher | Cat. #: 26156 | |
| Software, algorithm | Kaluza | Beckman Coulter | RRID:SCR_016182 | |
| Software, algorithm | GraphPad Prism software | GraphPad | RRID:SCR_002798 | |
| Other | anti-CD8 (Ly-2) MicroBeads | Miltenyi Biotec | Cat. #: 130-117-044 | |
| Other | dsDNA | Sigma | Cat. #: 31149 | ELISA (100 mg/mL) |

## Mice

C57BL/6 mice were obtained from the Vital River Laboratories (Beijing, China). *Cd11c-p28*^f/f mice (DC-specific deletion of p28) and *Il27ra*^-/- mice on C57BL/6 background were kindly provided by Dr. Zhinan Yin from Jinan University (Guangzhou, China). FVB-Tg (*Rag2*-EGFP) 1Mnz/J mice were purchased from Jackson Laboratory (Bar Harbor, ME) and were backcrossed for 10 generations onto the C57BL/6 background (termed as *Rag2*p-EGFP in this paper). *Rag2*p-EGFP mice were bred with *Cd11c-p28*^f/f mice to generate *Cd11c-p28*^f/f *Rag2*p-EGFP mice. *Aire*^-/- mice were generously provided by Yangxin Fu (University of Chicago, IL) and were bred with *Cd11c-p28*^f/f mice to generate *Aire*^-/- *Cd11c-p28*^f/f mice. Mice were used at 6–8 weeks of age unless stated otherwise. All the animal procedures were conformed to the Chinese Council on Animal Care Guidelines and the study was approved by the ethics committee of Peking University Health Science Center with an approval number of LA2014178.

## Antibodies and reagents

PE-Cy7-conjugated anti-mouse CD4 (RM4-5), PE- and APC-conjugated anti-mouse CD8a (53–6.7), APC-conjugated anti-mouse IL-2 (JES6-5H4), PE-Cy7-conjugated anti-mouse TNF-α (MP6-XT22), PE-conjugated anti-mouse Stat1 (pY701) (4 a), PerCP-Cy5.5-conjugated anti-mouse Stat3 (pY705) (4/P-STAT3), Alexa Fluor 488-conjugated anti-mouse Stat4 (pY693) (38/p-Stat4) were purchased from BD Biosciences (San Diego, CA). PE- and APC-conjugated anti-mouse CD25 (PC61.5) and CD44 (IM7), FITC-conjugated anti-mouse FR4 (eBio12A5), PerCP-eFluor710-conjugated anti-mouse CD73 (eBioTY/11.8), FITC-conjugated anti-mouse IL-4 (BVD6-24G2), PE-conjugated anti-mouse IL-17A (eBio17B7), APC-conjugated anti-mouse FOXP3 (3G3), and biotin-conjugated anti-mouse CD8a (53–6.7) were obtained from eBioscience (Waltham, MA). PE-conjugated anti-mouse IFN-γ (XMG1.2)

and PerCP-Cy5.5-conjugated anti-mouse T-bet (4B10) were purchased from BioLegend (San Diego, CA).

Functional grade monoclonal antibodies for murine CD3 (145–2 C11) and CD28 (37.51) and neutralizing anti-IL4 (11B11), neutralizing anti-IFN-γ (clone XMG1.2) were obtained from eBioscience (Waltham, MA). Recombinant murine IL-2, IL-4, IL-6, IL-12 and recombinant human TGF-β1 were purchased from R&D Systems (Abingdon, UK).

Anti-phospho-STAT1(Tyr701), anti-phospho-STAT-1(S727), anti-STAT1, anti-phospho-STAT3 (Tyr705), anti-STAT3, anti-phospho-STAT4 (Tyr693), anti-STAT4, anti-SOCS3, anti-Actin and HRP-labeled goat-anti-rabbit or anti-mouse IgGs for western blot assay were purchased from Cell Signaling Technology (Danvers, MA). Phorbol 12-myristate 13-acetate (PMA) and ionomycin were obtained from Sigma-Aldrich.

## Cell sorting and CD4[+] T-cell differentiation *in vitro*

To enrich CD4SP thymocytes, CD8[-] thymocytes were obtained by negative selection using anti-CD8 (Ly-2) MicroBeads (Miltenyi Biotec). The cells were then stained with fluorescently labeled antibodies to CD4, CD8 and CD44. CD4[+] SP thymocytes with the phenotype of GFP[+]CD4[+]CD8[-]CD44[lo] were then sorted. For the isolation of CD4[+] RTEs and mature naive CD4[+] T cells, GFP[+]CD4[+]CD8[-]CD25[-]NK1.1[-] (RTEs) and GFP[-]CD4[+]CD8[-]CD25[-]CD44[lo] (naive T) cells were sorted from lymph nodes. All these cells were sorted to >99% purity with a FACS Aria II (BD Biosciences, San Diego, CA).

For *in vitro* activation, CD4SP thymocytes, RTEs and mature naive T cells were cultured at a density of $2\times10^6$ /mL with plate-coated anti-CD3 (clone 2C11; 2 µg/mL) and soluble anti-CD28 (clone 37 N; 1 µg/mL) in RPMl 1640 medium supplemented with 10% fetal bovine serum (Biochrom Ag, Berlin), penicillin, streptomycin and 50 µM 2-mercaptoethanol. Conditions for CD4[+] T-cell differentiation were as follows: Th0 cells, IL-2 (2 ng/mL); Th1 cells, IL-12 (10 ng/mL), IL-2 (2 ng/mL), anti-IL-4 (10 µg/mL); Th2 cells, IL-4 (20 ng/mL), IL-2 (2 ng/mL), anti-IFN-γ (10 µg/mL); Th17 cells, recombinant human TGFβ1 (10 ng/mL), IL-6 (10 ng/mL), anti-IL-4 (10 µg/mL), anti-IFN-γ (10 µg/mL); and Treg cells, TGFβ1 (1 ng/mL), IL-2 (10 ng/mL), anti-IL-4 (10 µg/mL), anti-IFN-γ (10 µg/mL). After 3 days of differentiation, supernatants were collected and cells were subjected to intracellular cytokine staining and flow cytometry analyses.

## Intracellular staining and flow cytometry

For detection of surface molecules, T cells were labeled with the appropriate fluorescent mAbs on ice for 30 min. For detection of cytoplasmic molecules, T-bet, and FOXP3, T cells were collected, stained with surface molecules, fixed, permeabilized with the Foxp3/Transcription Factor Staining Buffer (eBioscience), and stained with fluorochrome-conjugated mouse antibodies on ice for 30 min. For intracellular cytokine staining, 5 hr before harvest, T cells were stimulated with PMA (50 ng/mL) plus ionomycin (1 mg/mL) in the presence of a protein transport inhibitor, monensin (2 µM) or BFA (3 µg/mL; eBioscience). Cells were collected, washed, fixed, permeabilized (FIX AND PERM, Invitrogen) and stained with IFN-γ, IL-4, IL-2, TNF-α, and IL-17 antibodies according to the manufacturer's instructions. For detection of phosphorylated STATs, $1\times10^7$ thymocytes were fixed for 10 min at 37 °C with 2% (wt/vol) paraformaldehyde. After fixing, cells were permeabilized for 30 min on ice with 90% (vol/vol) methanol, and stained with the appropriate antibodies. Flow cytometry was conducted on a Galios (Beckman Coulter) and data analysis was performed using Kaluza software.

## Enzyme-Linked Immunosorbent Assay (ELISA)

Supernatants of *in vitro* cell cultures were obtained at 72 hr. IFN-γ and IL-4 production were determined using ELISA kits (BioLegend, San Diego, CA) according to the manufacturer's instructions.

For the analysis of anti-dsDNA antibodies, serum was collected from 24- to 30-week-old *Aire[-/-] Cd11c-Cre p28[f/f]*, *Aire[-/-]*, *Cd11c-Cre p28[f/f]* and WT mice. Nunc MaxiSorp ELISA plates were precoated with dsDNA (100 mg/mL, Sigma, St. Louis, MO) in phosphate-buffered saline (PBS) at 4° C overnight. Plates were blocked with 5% BSA for 1 hr at 37° C, then washed and incubated with 1/100 dilutions of mouse sera for 2 hr at 37° C. Plates were washed, and anti-dsDNA antibodies were detected with a 1/1000 dilution of alkaline phosphatase–conjugated goat anti-mouse IgG (BioLegend) for 1 hr at 37° C and developed with a phosphatase substrate for 30 min at 37° C.

## Quantitative PCR (qPCR)

RNA was purified from various T cell subsets cultured under Th0 conditions for 12 hr using Trizol reagent (Invitrogen). cDNA was synthesized using reverse transcription kit (Progema). qPCR was performed using FastStart Universal SYBR Green Master mix (Roche, Basel, Switzerland) on an iCycler real-time PCR system (Bio-Rad Laboratories, Hemel Hempstead, U.K.), with each sample in triplicate. The quantification was based on delta delta CT calculations and was normalized to β-actin as loading controls. The primers used in the study were listed in *Supplementary file 1*.

## DNA methylation

Bisulfite modification of genomic DNA from FACS purified cells was performed using the EZ DNA methylation kit (Zymo Research). For methylation analysis on *Ifng* gene promoter CpG sites, bisulfite-treated DNA was amplified in semi-nested PCR using primers: 5'-GGTGTGAAGTAAAAGTGTTTTTAG AGAATTTTAT-3' and 5'-CAATAACAACCAAAAACAACCATAAAAAAAAACT-3', then 5'-GGTGTGAA GTAAAAGTGTTTTTAGAGAATTTTAT-3' and 5'-CCATAAAAAAAAAACTACAAAACCAAAATACAA TA-3'. For methylation analysis on *Il4* gene promoter CpG sites, bisulfite-treated DNA was amplified in PCR using primers: 5-GGATCCACACGGTGCAAAGAGAGACCC-3' and 5'-TCGGCCTTTCAGACTA ATCTTATCAGC-3' The PCR products were gel purified and cloned into the pGEM-T vector (Promega; Madison, WI, USA). The inserted PCR fragments of individual clones were sequenced by Tsing KE Biological Technology, Beijing, China. For all samples, 10 reads per CpG site were used to determine the average percentage of methylated CpG.

## Chromatin immunoprecipitation (ChIP)

ChIP assays were carried out by using Pierce Agarose Chip Kit (Pierce Biotechnology). Briefly, cells were cross-linked for 10 min with 1% formaldehyde and lysed with Nuclease. 10% of the digested chromatin was preserved as input control and the rest of the digested chromatin were incubated with specific antibody or normal IgG (as a control). The purified DNA from immunoprecipitation and the input samples were analyzed by qPCR using primers specific listed in *Supplementary file 1*. Data is presented as a percent input of each IP sample relative to input chromatin. The following antibodies were used in ChIP analysis: H3K4me3 (Millipore), H3K27me3 (Millipore), H3K9me3 (Abcam), and STAT1 (Cell Signaling Technology).

## Western blotting

For short-term TCR stimulation, CD4SP thymocytes and mature naive T cells were incubated with 2 µg/mL anti-mouse CD3 and 1 µg/mL anti-mouse CD28 on ice for 20 min, followed by cross-linking for 10 min at 37 °C with 5 µg/mL goat-anti-hamster IgG. Cells were then washed with PBS. Freshly isolated T cells or stimulated T cells were lysed in RIPA buffer for 30 min. Whole-cell lysates (5–10 µg per sample) were separated by SDS/PAGE and analyzed by immunoblotting with antibodies to phospho-STAT-1 (Tyr701), phospho-STAT-1 (Ser727), STAT-1, phospho-STAT-3 (Tyr705), STAT-3, phospho-STAT-4 (Tyr693), STAT-4, SOCS3 and Actin. HRP-conjugated anti-Rabbit IgG was used as the detection antibody. The bands were quantified using ImageJ software.

## RNA-seq

Total RNA was extracted from approximately $2 \times 10^6$ CD4SP thymocytes from WT and *Cd11c-p28*[f/f] mice using Trizol reagent (Invitrogen). Each sample was a mixture of equal amounts of CD4SP thymocytes from three mice. RNA underwent quality control testing using a bionalayser followed by cDNA library preparation. Library construction and sequencing on Illumina HiSEq 2000 were performed by BIOPIC, Peking University. The analyzed FPKM values are provided in the Supplementary Materials (*Supplementary file 2*).

## H&E

Organs from 24- to 30-week-old *Aire*[-/-]*Cd11c-p28*[f/f] mice, *Aire*[-/-] mice and WT mice were collected and fixed overnight in 10% formalin, embedded in paraffin, sectioned and stained with hematoxylin and eosin (H&E). The degree of lymphocytic infiltrates was analyzed in a blinded fashion. In general, 0, 1, 2, and 3 indicate no, mild, moderate, or severe lymphocytic infiltration, respectively.

## Statistical analysis

Data are reported as the mean ± SD. Differences between groups were analyzed by Student's *t* test or two-way ANOVA (for multiple variant comparisons) using GraphPad Prism software (GraphPad). Throughout the text, figures, and figure legends, the following terminology is used to denote statistical significance: *, $p<0.05$; **, $p<0.01$; ***, $p<0.001$; NS, not significant.

## Acknowledgements

We thank Dr. Zhinan Yin from Jinan University (Guangzhou, China) for kindly provided *Cd11c-p28*flox/flox mice and *Il27ra*-/- mice. This work was supported by grants from the National Natural Sciences Foundation of China (32230037; 82394412; 32071178; 31872733), National Key Research and Development Program of China (2023YFB3507000) and Beijing Life Science Academy (2024600CC0080).

## Additional information

### Funding

| Funder | Grant reference number | Author |
|---|---|---|
| National Natural Science Foundation of China | 32230037 | Yu Zhang |
| National Natural Science Foundation of China | 82394412 | Yu Zhang |
| National Natural Science Foundation of China | 32071178 | Rong Jin |
| National Natural Science Foundation of China | 31872733 | Rong Jin |
| National Key Research and Development Program of China | 2023YFB3507000 | Rong Jin |
| Beijing Life Science Academy | 2024600CC0080 | Rong Jin |

The funders had no role in study design, data collection and interpretation, or the decision to submit the work for publication.

### Author contributions

Jie Zhang, Data curation, Investigation, Methodology, Writing – original draft, Project administration; Hui Tang, Data curation, Investigation, Methodology, Project administration; Haoming Wu, Software, Methodology; Xuewen Pang, Methodology; Rong Jin, Yu Zhang, Conceptualization, Supervision, Funding acquisition, Writing – review and editing

### Author ORCIDs

Jie Zhang ⓘ https://orcid.org/0000-0003-3338-8139
Rong Jin ⓘ https://orcid.org/0000-0002-5666-9497

### Ethics

All the animal procedures were conformed to the Chinese Council on Animal Care Guidelines and the study was approved by the ethics committee of Peking University Health Science Center with an approval number of LA2014178.

Reviewer #1 (Public Review): https://doi.org/10.7554/eLife.96868.3.sa1
Reviewer #2 (Public Review): https://doi.org/10.7554/eLife.96868.3.sa2
Author response https://doi.org/10.7554/eLife.96868.3.sa3

## Additional files

### Supplementary files

Supplementary file 1. The primers used in the study.

Supplementary file 2. The analyzed FPKM values of CD4SP thymocytes from *Cd11c-p28*$^{f/f}$ and WT mice.

MDAR checklist

### Data availability

All data generated or analyzed during this study are included in the manuscript and supporting files; source data files have been provided for Figures 1, 3 and 5.

The following previously published dataset was used:

| Author(s) | Year | Dataset title | Dataset URL | Database and Identifier |
|---|---|---|---|---|
| Hirahara K, Onodera A, Villarino AV, Bonelli M, Sciumè G, Laurence A, O'Shea JJ | 2015 | Asymmetric Action of STAT Transcription Factors Drives Transcriptional Outputs and Cytokine Specificity | https://www.ncbi.nlm.nih.gov/geo/query/acc.cgi?acc=GSE65621 | NCBI Gene Expression Omnibus, GSE65621 |

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
