## [Editor Report · eLife Assessment]

This study presents a **useful** reassessment of the potential role of dendritic cell-derived IL-27 p28 cytokine in the functional maturation of CD4+CD8- thymocytes, and CD4+ recent thymic emigrants. The evidence supporting the claims of the authors is **solid** and serves to reaffirm what has been previously described, with the overall advance in understanding the mechanism(s) responsible for the intrathymic functional programming of CD4+ T cells being limited.

---

## [Referee Report · Reviewer #1 (Public Review)]

Summary:

Zhang et al. demonstrate that CD4+ single positive (SP) thymocytes, CD4+ recent thymic emigrants (RTE), and CD4+ T naive (Tn) cells from Cd11c-p28-flox mice, which lack IL-27p28 selectively in Cd11c+ cells, exhibit a hyper-Th1 phenotype instead of the expected hyper Th2 phenotype. Using IL-27R-deficient mice, the authors confirm that this hyper-Th1 phenotype is due to IL-27 signaling via IL-27R, rather than the effects of monomeric IL-27p28. They also crossed Cd11c-p28-flox mice with autoimmune-prone Aire-deficient mice and showed that both T cell responses and tissue pathology are enhanced, suggesting that SP, RTE, and Tn cells from Cd11c-p28-flox mice are poised to become Th1 cells in response to self-antigens. Regarding mechanism, the authors demonstrate that SP, RTE, and Tn cells from Cd11c-p28-flox mice have reduced DNA methylation at the IFN-g and Tbx21 loci, indicating 'de-repression', along with enhanced histone tri-methylation at H3K4, indicating a 'permissive' transcriptional state. They also find evidence for enhanced STAT1 activity, which is relevant given the well-established role of STAT1 in promoting Th1 responses, and surprising given IL-27 is a potent STAT1 activator. This latter finding suggests that the Th1-inhibiting property of thymic IL-27 may not be due to direct effects on the T cells themselves.

Strengths:

Overall the data presented are high quality and the manuscript is well-reasoned and composed. The basic finding - that thymic IL-27 production limits the Th1 potential of SP, RTE, and Tn cells - is both unexpected and well described.

Weaknesses from the original round of review:

A credible mechanistic explanation, cellular or molecular, is lacking. The authors convincingly affirm the hyper-Th1 phenotype at epigenetic level but it remains unclear whether the observed changes reflect the capacity of IL-27 to directly elicit epigenetic remodeling in developing thymocytes or knock-on effects from other cell types which, in turn, elicit the epigenetic changes (presumably via cytokines). The authors propose that increased STAT1 activity is a driving force for the epigenetic changes and resultant hyper-Th1 phenotype. That conclusion is logical given the data at hand but the alternative hypothesis - that the hyper-STAT1 response is just a downstream consequence of the hyper-Th1 phenotype - remains equally likely. Thus, while the discovery of a new anti-inflammatory function for IL-27 within the thymus is compelling, further mechanistic studies are needed to advance the finding beyond phenomenology.

---

## [Referee Report · Reviewer #2 (Public Review)]

Summary:

Naïve CD4 T cells in CD11c-Cre p28-floxed mice express highly elevated levels of proinflammatory IFNg and the transcription factor T-bet. This phenotype turned out to be imposed by thymic dendritic cells (DCs) during CD4SP T cell development in the thymus [PMID: 23175475]. The current study affirms these observations, first, by developmentally mapping the IFNg dysregulation to newly generated thymic CD4SP cells [PMID: 23175475], second, by demonstrating increased STAT1 activation being associated with increased T-bet expression in CD11c-Cre p28-floxed CD4 T cells [PMID: 36109504], and lastly, by confirming IL-27 as the key cytokine in this process [PMID: 27469302]. The authors further demonstrate that such dysregulated cytokine expression is specific to the Th1 cytokine IFNg, without affecting the expression of the Th2 cytokine IL-4, thus proposing a role for thymic DC-derived p28 in shaping the cytokine response of newly generated CD4 helper T cells. Mechanistically, CD4SP cells of CD11c-Cre p28-floxed mice were found to display epigenetic changes in the Ifng and Tbx21 gene loci that were consistent with increased transcriptional activities of IFNg and T-bet mRNA expression. Moreover, in autoimmune Aire-deficiency settings, CD11c-Cre p28-floxed CD4 T cells still expressed significantly increased amounts of IFNg, exacerbating the autoimmune response and disease severity. Based on these results, the investigators propose a model where thymic DC-derived IL-27 is necessary to suppress IFNg expression by CD4SP cells and thus would impose a Th2-skewed predisposition of newly generated CD4 T cells in the thymus, potentially relevant in autoimmunity.

Strengths:

Experiments are well-designed and executed. The conclusions are convincing and supported by the experimental results.

Weaknesses from the original round of review:

The premise of the current study is confusing as it tries to use the CD11c-p28 floxed mouse model to explain the Th2-prone immune profile of newly generated CD4SP thymocytes. Instead, it would be more helpful to (1) give full credit to the original study which already described the proinflammatory IFNg+ phenotype of CD4 T cells in CD11c-p28 floxed mice to be mediated by thymic dendritic cells [PMID: 23175475], and then, (2) build on that to explain that this study is aimed to understand the molecular basis of the original finding.

In its essence, this study mostly rediscovers and reaffirms previously reported findings, but with different tools. While the mapping of epigenetic changes in the IFNg and T-bet gene loci and the STAT1 gene signature in CD4SP cells are interesting, these are expected results, and they only reaffirm what would be assumed from the literature. Thus, there is only incremental gain in new insights and information on the role of DC-derived IL-27 in driving the Th1 phenotype of CD4SP cells in CD11c-p28 floxed mice.

Altogether, the major issues of this study remain unresolved:

(1) It is still unclear why the p28-deficiency in thymic dendritic cells would result in increased STAT1 activation in CD4SP cells. Based on their in vitro experiments with blocking anti-IFNg antibodies, the authors conclude that it is unlikely that the constitutive activation of STAT1 would be a secondary effect due to autocrine IFNg production by CD4SP cells. However, this possibility should be further tested with in vivo models, such as Ifng-deficient CD11c-p28 floxed mice. Alternatively, is this an indirect effect by other IFNg producers in the thymus, such as iNKT cells? It is necessary to explain what drives the STAT1 activation in CD11c-p28 floxed CD4SP cells in the first place.

(2) It is also unclear whether CD4SP cells are the direct targets of IL-27 p28. The cell-intrinsic effects of IL-27 p28 signaling in CD4SP cells should be assessed and demonstrated, ideally by CD4SP-specific deletion of IL-27Ra, or by establishing bone marrow chimeras of IL-27Ra germline KO mice.

[Editors' note: The resubmitted paper was minimally revised, and many of the initial concerns remain unresolved.]

---

## [Author Response]

The following is the authors’ response to the original reviews

**Public Reviews:**

**Reviewer #1 (Public Review):**
Summary:Zhang et al. demonstrate that CD4^+^ single positive (SP) thymocytes, CD4^+^ recent thymic emigrants (RTE), and CD4^+^ T naive (Tn) cells from Cd11c-p28-flox mice, which lack IL-27p28 selectively in Cd11c+ cells, exhibit a hyper-Th1 phenotype instead of the expected hyper Th2 phenotype. Using IL-27R-deficient mice, the authors confirm that this hyper-Th1 phenotype is due to IL-27 signaling via IL-27R, rather than the effects of monomeric IL-27p28. They also crossed Cd11c-p28-flox mice with autoimmune-prone Aire-deficient mice and showed that both T cell responses and tissue pathology are enhanced, suggesting that SP, RTE, and Tn cells from Cd11c-p28-flox mice are poised to become Th1 cells in response to self-antigens. Regarding mechanism, the authors demonstrate that SP, RTE, and Tn cells from Cd11c-p28-flox mice have reduced DNA methylation at the IFN-g and Tbx21 loci, indicating 'de-repression', along with enhanced histone tri-methylation at H3K4, indicating a 'permissive' transcriptional state. They also find evidence for enhanced STAT1 activity, which is relevant given the well-established role of STAT1 in promoting Th1 responses, and surprising given IL-27 is a potent STAT1 activator. This latter finding suggests that the Th1-inhibiting property of thymic IL-27 may not be due to direct effects on the T cells themselves.Strengths:Overall the data presented are high quality and the manuscript is well-reasoned and composed. The basic finding - that thymic IL-27 production limits the Th1 potential of SP, RTE, and Tn cells - is both unexpected and well described.Weaknesses:A credible mechanistic explanation, cellular or molecular, is lacking. The authors convincingly affirm the hyper-Th1 phenotype at epigenetic level but it remains unclear whether the observed changes reflect the capacity of IL-27 to directly elicit epigenetic remodeling in developing thymocytes or knock-on effects from other cell types which, in turn, elicit the epigenetic changes (presumably via cytokines). The authors propose that increased STAT1 activity is a driving force for the epigenetic changes and resultant hyper-Th1 phenotype. That conclusion is logical given the data at hand but the alternative hypothesis - that the hyper-STAT1 response is just a downstream consequence of the hyper-Th1 phenotype - remains equally likely. Thus, while the discovery of a new anti-inflammatory function for IL-27 within the thymus is compelling, further mechanistic studies are needed to advance the finding beyond phenomenology.

Thank you for your insightful comments and suggestions. We appreciate your feedback and have carefully considered the concerns raised regarding the mechanistic explanation of our findings. To address the issue of whether developing thymocytes are the direct targets of IL-27, we plan to conduct further studies using *Cd4-IL-27ra* knockout mice or mixed bone marrow chimeras consisting of wildtype and *IL-27ra* knockout cells. This approach will help us determine if IL-27 directly induces epigenetic remodeling in thymocytes or if the observed effects are secondary to influences from other cell types.

Regarding the potential autocrine loop contributing to STAT1 hyperactivation, we have performed preliminary experiments by adding IFN-γ antibody to CD4^+^ T cell cultures and observed no significant impact on STAT1 phosphorylation. If necessary, we will further investigate this possibility in vivo using *Cd4-Ifng* and *CD11c-p28* double knockout mice.

The detailed mechanisms underlying STAT1 hyperactivation remain to be elucidated. Recent studies have shown that IL-27p28 can act as an antagonist of gp130-mediated signaling. Structural analyses have also demonstrated that IL-27p28 interacts with EBI3 and the two receptor subunits IL-27Rα and gp130. Given these findings and the similar phenotypes observed in p28 and IL-27ra deficient mice, we speculate that the deficiency of either p28 or IL-27ra may increase the availability of gp130 for signaling by other cytokines. We will focus our future research on gp130-related cytokines to identify potential candidates that could lead to enhanced STAT1 activation in the absence of p28. Alternatively, the release of EBI3 in p28-deficient conditions may promote its interaction with other cytokine subunits. IL-35, which is composed of EBI3 and p35, is of particular interest given the involvement of IL-27Rα in its signaling pathway.

To narrow down the candidate cytokines, we reanalyzed single-cell RNA sequencing data from *CD11c-cre p28f/f* and wild-type thymocytes (Signal Transduct Target Ther. 2022, DOI: 10.1038/s41392-022-01147-z). Our analysis revealed that thymic dendritic cells (DCs) were categorized into two distinct clusters, with both *Il12a* (p35, which forms IL-35 with EBI3) and *Clcf1* (CLCF1) being upregulated in *CD11c-cre p28f/f* mice. In CD4 single-positive (SP) thymocytes, the expression levels of gp130 and IL-12Rβ2 (the receptor for IL-35) were comparable between knockout and wild-type mice. However, the mRNA levels of *Lifr* and *Cntfr* were low in CD4 SP thymocytes.

**Author response image 1. sa3fig1:** Single-cell RNA sequencing data from CD11c-cre *p28f/f* (KO) and wild-type thymocytes (Signal Transduct Target Ther. 2022, DOI: 10.1038/s41392-022-01147-z).

We have planned to assess the protein levels of IL-35 and CLCF1 in dendritic cells, as well as their respective receptors, to evaluate their effects on STAT1 phosphorylation in CD4^+^ thymocytes from both wild-type and p28-deficient mice. Unfortunately, we have encountered challenges with the mouse breeding and anticipate that it will take approximately six months to obtain the appropriate genotype necessary to complete these experiments.

**Reviewer #2 (Public Review):**
Summary:Naïve CD4 T cells in CD11c-Cre p28-floxed mice express highly elevated levels of proinflammatory IFNg and the transcription factor T-bet. This phenotype turned out to be imposed by thymic dendritic cells (DCs) during CD4SP T cell development in the thymus [PMID: 23175475]. The current study affirms these observations, first, by developmentally mapping the IFNg dysregulation to newly generated thymic CD4SP cells [PMID: 23175475], second, by demonstrating increased STAT1 activation being associated with increased T-bet expression in CD11c-Cre p28-floxed CD4 T cells [PMID: 36109504], and lastly, by confirming IL-27 as the key cytokine in this process [PMID: 27469302]. The authors further demonstrate that such dysregulated cytokine expression is specific to the Th1 cytokine IFNg, without affecting the expression of the Th2 cytokine IL-4, thus proposing a role for thymic DC-derived p28 in shaping the cytokine response of newly generated CD4 helper T cells. Mechanistically, CD4SP cells of CD11c-Cre p28-floxed mice were found to display epigenetic changes in the Ifng and Tbx21 gene loci that were consistent with increased transcriptional activities of IFNg and T-bet mRNA expression. Moreover, in autoimmune Aire-deficiency settings, CD11c-Cre p28-floxed CD4 T cells still expressed significantly increased amounts of IFNg, exacerbating the autoimmune response and disease severity. Based on these results, the investigators propose a model where thymic DC-derived IL-27 is necessary to suppress IFNg expression by CD4SP cells and thus would impose a Th2-skewed predisposition of newly generated CD4 T cells in the thymus, potentially relevant in autoimmunity.Strengths:Experiments are well-designed and executed. The conclusions are convincing and supported by the experimental results.Weaknesses:The premise of the current study is confusing as it tries to use the CD11c-p28 floxed mouse model to explain the Th2-prone immune profile of newly generated CD4SP thymocytes. Instead, it would be more helpful to (1) give full credit to the original study which already described the proinflammatory IFNg+ phenotype of CD4 T cells in CD11c-p28 floxed mice to be mediated by thymic dendritic cells [PMID: 23175475], and then, (2) build on that to explain that this study is aimed to understand the molecular basis of the original finding.In its essence, this study mostly rediscovers and reaffirms previously reported findings, but with different tools. While the mapping of epigenetic changes in the IFNg and T-bet gene loci and the STAT1 gene signature in CD4SP cells are interesting, these are expected results, and they only reaffirm what would be assumed from the literature. Thus, there is only incremental gain in new insights and information on the role of DC-derived IL-27 in driving the Th1 phenotype of CD4SP cells in CD11c-p28 floxed mice.

Thank you for your valuable comments and suggestions. We appreciate your input and have carefully reviewed the concerns raised regarding the premise and novelty of our study.

Indeed, the current study is built upon the foundational work of Zhang et al. (PMID: 23175475), which first described the proinflammatory IFN-γ^+^ phenotype of CD4 T cells in CD11c-p28 floxed mice mediated by thymic dendritic cells. We have cited this study multiple times in our manuscript to acknowledge its significance. Our goal was to expand on this original finding by exploring the functional bias of newly generated CD4^+^ T cells, elucidating the mechanisms underlying the hyper-Th1 phenotype in the absence of thymic DC-derived IL-27, and exploring its relevance in pathogenesis of autoimmunity.

Our study revisits this phenomenon with a focus on the molecular and epigenetic changes that drive the Th1 bias in CD4SP cells. We demonstrated that the deletion of p28 in thymic dendritic cells leads to an unexpected hyperactivation of STAT1, which is associated with epigenetic modifications that favor Th1 differentiation. These findings provide a deeper understanding of the molecular basis behind the original observation of the Th1-skewed phenotype in CD11c-p28 floxed mice.

However, as you pointed out, there is still a gap in understanding the precise link between p28 deficiency and STAT1 activation. We acknowledge that our study primarily reaffirms previously reported findings with different tools and approaches. While the mapping of epigenetic changes in the IFN-γ and T-bet gene loci and the STAT1 gene signature in CD4SP cells are interesting, they are indeed expected results based on the existing literature. This limits the novelty and incremental gain in new insights provided by our study.

To address this gap and enhance the novelty of our findings, we plan to conduct further investigations to elucidate the detailed mechanisms connecting p28 deficiency to STAT1 hyperactivation. We will explore potential compensatory pathways or alternative signaling mechanisms that may contribute to the observed epigenetic changes and Th1 bias. Additionally, we will consider the broader impact of IL-27 deficiency on the thymic environment and its downstream effects on CD4^+^ T cell differentiation.

We appreciate your feedback and will work to strengthen the mechanistic underpinnings of our study. We believe that these additional efforts will provide a more comprehensive understanding of the role of DC-derived IL-27 in shaping the Th1 phenotype of CD4SP cells and contribute meaningful insights to the field.

Altogether, the major issues of this study remain unresolved:(1) It is still unclear why the p28-deficiency in thymic dendritic cells would result in increased STAT1 activation in CD4SP cells. Based on their in vitro experiments with blocking anti-IFNg antibodies, the authors conclude that it is unlikely that the constitutive activation of STAT1 would be a secondary effect due to autocrine IFNg production by CD4SP cells. However, this possibility should be further tested with in vivo models, such as Ifng-deficient CD11c-p28 floxed mice. Alternatively, is this an indirect effect by other IFNg producers in the thymus, such as iNKT cells? It is necessary to explain what drives the STAT1 activation in CD11c-p28 floxed CD4SP cells in the first place.

Thank you for your insightful suggestions. We appreciate your feedback and are committed to addressing the critical questions raised regarding the mechanisms underlying STAT1 activation in CD4SP cells in the context of p28 deficiency in thymic dendritic cells.

To further investigate the potential autocrine loop for IFN-γ production, we will conduct in vivo studies using Cd4-Ifng and CD11c-p28 double knockout mice. This model will allow us to directly test whether IFN-γ produced by CD4SP cells themselves contributes to the observed STAT1 activation. Additionally, this approach will help exclude the possibility of indirect effects from other IFN-γ-producing cells in the thymus, such as invariant natural killer T (iNKT) cells, as suggested by the reviewer.

As you correctly pointed out, a key unanswered question is what drives the initial STAT1 activation in CD4SP cells of CD11c-p28 floxed mice. Our current hypothesis is that p28 deficiency enhances the responsiveness of developing thymocytes to STAT1-activating cytokines. This hypothesis is supported by several lines of evidence:

(1) Functional Antagonism: Recent studies have shown that IL-27p28 can act as an antagonist of gp130-mediated signaling. This suggests that in the absence of p28, the inhibitory effect of IL-27p28 on downstream signaling may be lost, leading to increased sensitivity to other cytokines that activate STAT1.

(2) Structural Insights: Structural studies have demonstrated that IL-27p28 is centrally positioned within the complex formed with EBI3 and the two receptor subunits IL-27Rα and gp130. This positioning implies that p28 deficiency could disrupt the balance of cytokine signaling pathways involving these components.

(3) Phenotypic Similarity: We have observed a similar hyper-Th1 phenotype in mice lacking either p28 or IL-27ra. This similarity suggests that the absence of p28 may lead to increased availability of gp130 for signaling by other cytokines, thereby enhancing STAT1 activation.

Based on these considerations, we hypothesize that the deficiency of p28 results in a greater availability of gp130 to transduce signals from other cytokines, ultimately leading to enhanced STAT1 activation in CD4SP cells. To identify the specific cytokine(s) responsible for this effect, we will focus on gp130-related cytokines, as outlined in our response to Reviewer 1. This will involve reanalysis of single-cell RNA sequencing data and further experimental validation to pinpoint the candidate cytokines driving the observed STAT1 hyperactivation.

We are confident that these additional studies will provide a clearer understanding of the mechanisms linking p28 deficiency in thymic dendritic cells to increased STAT1 activation in CD4SP cells. We appreciate your guidance and look forward to sharing our findings.

(2) It is also unclear whether CD4SP cells are the direct targets of IL-27 p28. The cell-intrinsic effects of IL-27 p28 signaling in CD4SP cells should be assessed and demonstrated, ideally by CD4SP-specific deletion of IL-27Ra, or by establishing bone marrow chimeras of IL-27Ra germline KO mice.

Thanks for the suggestions. Further studies will be performed to test whether developing thymocytes are the direct targets of IL-27 using *Cd4-IL-27ra* knockout mice or mixed bone marrow chimeras of wildtype and *IL-27ra* knockout cells. Unfortunately, we have encountered challenges with the mouse breeding and anticipate that it will take approximately six months to obtain the appropriate genotype necessary to complete these experiments.

**Recommendations for the authors:**

**Reviewer #1 (Recommendations For The Authors):**
(1) Is the hyper-STAT1 response seen in T cells from Cd11c-p28-flox mice due to increased availability and/or increased responsiveness to STAT1 activating cytokines? Studies, where SP, RTE, and Tn cells are pulsed ex vivo with IL-27 and/or other STAT1-activating cytokines, would address the latter (with STAT1 phosphorylation as the major readout). Given the ability of IL-27 to activate STAT3, this pathway should also be addressed. It would be of interest if STAT1 signaling is selectively impaired, as suggested by the work of Twohig et al. (doi: 10.1038/s41590-019-0350-0.)(which should be cited and discussed).

Thank you for your insightful suggestions. We appreciate your input and are committed to addressing the critical questions raised regarding the mechanisms underlying the hyper-activation of STAT1 in T cells from Cd11c-p28-flox mice.

The detailed mechanisms driving the hyper-activation of STAT1 remain to be fully elucidated. Recent studies have shown that IL-27p28 can act as an antagonist of gp130-mediated signaling. Structural analyses have also demonstrated that IL-27p28 interacts with EBI3 and the two receptor subunits IL-27Rα and gp130. Considering these findings and the similar phenotypes observed in p28 and IL-27ra deficient mice, we speculate that the deficiency of either p28 or IL-27ra may increase the availability of gp130 for signaling by other cytokines. This could potentially enhance the responsiveness of developing thymocytes to STAT1-activating cytokines. We will focus our future research on gp130-related cytokines to identify the candidate(s) responsible for the enhanced STAT1 activation in the absence of p28. Alternatively, the release of EBI3 in the absence of p28 may facilitate its coupling with other cytokine subunits. IL-35, which is composed of EBI3 and p35, is of particular interest given the involvement of IL-27Rα in its signaling pathway.

To narrow down the candidate cytokines, we reanalyzed single-cell RNA sequencing data from CD11c-cre *p28f/f* and wild-type thymocytes (Signal Transduct Target Ther. 2022, DOI: 10.1038/s41392-022-01147-z). Our analysis revealed that thymic dendritic cells (DCs) were categorized into two distinct clusters, with both *Il12a* (p35, which forms IL-35 with EBI3) and *Clcf1* (CLCF1) being upregulated in *CD11c-cre p28f/f* mice. In CD4 single-positive (SP) thymocytes, the expression levels of gp130 and IL-12Rβ2 (the receptor for IL-35) were comparable between knockout and wild-type mice. However, the mRNA levels of *Lifr* and *Cntfr* were low in CD4 SP thymocytes.

Single-cell RNA sequencing data from CD11c-cre *p28f/f* (KO) and wild-type thymocytes (Signal Transduct Target Ther. 2022, DOI: 10.1038/s41392-022-01147-z).

We have planned to assess the protein levels of IL-35 and CLCF1 in dendritic cells, as well as their respective receptors, to evaluate their effects on STAT1 phosphorylation in CD4^+^ thymocytes from both wild-type and p28-deficient mice. Unfortunately, we have encountered challenges with the mouse crosses and anticipate that it will take approximately six months to obtain the appropriate genotype necessary to complete these experiments.

As you correctly noted, the ability of IL-27 to activate STAT3 signaling is an important consideration. We have carefully examined this pathway in our current study, and our results indicate that neither total nor phosphorylated STAT3 and STAT4 were found to be altered with IL-27p28 ablation (Figure 5B). This suggests that the impact is indeed specific to the STAT1 axis. We will also consider the possibility of selective impairment of STAT1 signaling, as suggested by the work of Twohig et al. (doi: 10.1038/s41590-019-0350-0), which we will cite and discuss in our revised manuscript.

We appreciate your guidance and will work diligently to address these questions in our future studies. We look forward to sharing our findings and contributing to a deeper understanding of the role of IL-27 in the regulation of STAT1 activation in T cells.

(2) It may be that the hyper-Th1 phenotype is not due to cell-intrinsic differences in STAT1 signaling (see Major Point 1) but rather, hyper-responsiveness to TCR + Co-stimulation (as provided in the re-stim assays used throughout). This issue is particularly relevant for the ChIP studies where the author notes that, "...we chose to treat the cells with anti-CD3 and anti-CD28 for 3 days prior to the assay". Why not treat these cells ex vivo with STAT1-activating cytokines instead of anti-CD3/CD28? The current methodology makes it impossible to distinguish between enhanced TCR/CD28 and cytokine signaling, and ultimately does not address SP, RTE, and Tn cells (since they are now activated, blasts.).

Thank you for raising this important point. We appreciate your feedback and fully recognize the limitations of our current methodology, which uses anti-CD3/CD28 stimulation for ChIP experiments. This approach indeed complicates the distinction between enhanced TCR/CD28 signaling and cytokine-mediated STAT1 activation, particularly in the context of SP, RTE, and Tn cells, which become activated blasts under these conditions.

To address these concerns and provide more precise insights into the mechanisms underlying the hyper-Th1 phenotype, we are revising our experimental strategy. Specifically, we are shifting our focus to directly investigate the role of STAT1-activating cytokines in the absence of p28. Based on our previous analysis and re-evaluation of single-cell RNA sequencing data, we have identified IL-35 and CLCF1 as the most promising candidate cytokines.

We are now planning to perform ChIP experiments using these cytokines directly, rather than relying on TCR + co-stimulation. This approach will allow us to more accurately evaluate the impact of these cytokines on STAT1 signaling in CD4^+^ T cells. By treating cells *ex vivo* with IL-35 and CLCF1, we aim to elucidate whether the observed hyper-Th1 phenotype is driven by enhanced responsiveness to these cytokines, independent of TCR/CD28 signaling.

We regret to inform you that we have encountered unforeseen challenges with mouse crosses, which have delayed our progress. As a result, we anticipate a delay of approximately six months to obtain the appropriate genotypes necessary to complete these experiments. We understand the importance of these revisions and are committed to overcoming these challenges to provide a more robust and accurate analysis.

(3) Studies involving STAT1-deficient mice are necessary (ideally with STAT1 deficiency restricted to the T cell compartment). At a minimum, it must be confirmed that these phenocopy Cd11c-p28-flox mice in terms of SP, RTE, and Tn cells (and their Th1-like character). If a similar hyper-Th1 phenotype is not seen, then the attendant hyper STAT1 response can only be viewed as a red herring.

Thank you for raising this important consideration. We acknowledge the significance of addressing the role of STAT1 specifically within the T cell compartment to validate the mechanisms underlying the hyper-Th1 phenotype observed in Cd11c-p28-flox mice.

We agree that studies involving STAT1-deficient mice, particularly with STAT1 deficiency restricted to the T cell compartment, are essential to confirm whether the hyper-Th1 phenotype is directly driven by STAT1 hyperactivation in T cells. Ideally, such studies would help determine if STAT1 deficiency in T cells phenocopies the Cd11c-p28-flox mice, particularly in terms of the SP, RTE, and Tn cells and their Th1-like characteristics.

Unfortunately, we currently face challenges in obtaining and breeding the appropriate STAT1 conditional knockout mice with T cell-specific deletion. This has limited our ability to conduct these experiments in a timely manner. However, we recognize the importance of these studies and are actively working to secure the necessary resources and models to address this critical question.

We understand that without these experiments, any conclusions drawn about the role of STAT1 hyperactivation in driving the hyper-Th1 phenotype must be considered with caution. If a similar hyper-Th1 phenotype is not observed in STAT1-deficient T cells, then the hyper-STAT1 response may indeed be a secondary or compensatory effect rather than a primary driver.

We are committed to pursuing these studies and will prioritize them in our future work. We will keep you informed of our progress and will update the manuscript with the results of these experiments once completed. We appreciate your patience and understanding as we work to address this important aspect of our research.

(4) The authors mine their RNA-seq data using a STAT1 geneset sourced from studies involving IL-21 as the upstream stimulus. Why was this geneset was chosen? It is true that IL-21 can activate STAT1 but STAT3 is typically viewed as its principal signaling pathway. There are many more appropriate genesets, especially from studies where T cells are cultured with traditional STAT1 stimuli (e.g. IL-27 in Hirahara et al., Immunity 2015 or interferons in Iwata et al., Immunity 2017)doi: 10.1016/j.immuni.2015.04.014, 10.1016/j.immuni.2017.05.005.

Thank you for your insightful comments. We appreciate your attention to the choice of the STAT1 gene set in our RNA-seq analysis.

Initially, we selected the STAT1 gene set from a study involving IL-21 stimulation (GSE63204) because IL-21 is known to activate STAT1, despite STAT3 being its principal signaling pathway. However, we acknowledge that this choice may not have been optimal given the context of our study, which focuses on the role of IL-27 and its impact on STAT1 signaling in T cells.

We agree that gene sets derived from studies using more canonical STAT1 stimuli, such as IL-27 or interferons, would be more relevant for our analysis. In response to your suggestion, we have revised our approach and adopted a gene set from GSE65621, which compares STAT1-/- and wild-type CD4 T cells following IL-27 stimulation. This gene set is more aligned with the focus of our study and provides a more appropriate reference for identifying STAT1-activated genes.

Our re-analysis revealed that 270 genes (FPKM > 1, log2FC > 2) were downregulated in STAT1-/- cells compared to wild-type cells, which we defined as STAT1-activated genes. Notably, approximately 50% of the upregulated differentially expressed genes (55 out of 137) in our dataset fell into the category of STAT1-activated genes, while none were classified as STAT1-suppressed genes (Figure 4B). Furthermore, Gene Set Enrichment Analysis (GSEA) demonstrated significant enrichment of STAT1-activated genes in the transcriptome of CD4 SP thymocytes from the knockout mice (NES = 1.67, nominal p-value = 10^-16^, Figure 4D).

These findings support our conclusion that IL-27p28 deficiency leads to enhanced STAT1 activity in CD4 SP thymocytes. We believe that using a more relevant gene set has strengthened our analysis and provided clearer insights into the molecular mechanisms underlying the observed phenotype.

We have cited the relevant studies (Hirahara et al., Immunity 2015; Iwata et al., Immunity 2017) to provide context for our revised analysis and to acknowledge the importance of canonical STAT1 stimuli in T cell signaling. We appreciate your guidance and are confident that these revisions have improved the robustness and relevance of our findings.

(5) Given the ability of IL-27 to activate STAT1 in T cells, it is surprising that SP, RTE, and Tn cells from Cd11c-p28-flox mice exhibit more STAT1 signaling than WT controls. If not IL-27, then what is the stimulus for this STAT1 activity? The authors rule out autocrine IFN-g production in vitro (not in vivo) but provide no further insight.

Thank you for raising this important question. We appreciate your interest in understanding the source of enhanced STAT1 signaling in SP, RTE, and Tn cells from Cd11c-p28-flox mice, especially given the role of IL-27 in activating STAT1 in T cells. As previously discussed, we have identified IL-35 and CLCF1 as the most likely candidate cytokines driving the observed STAT1 activity in the absence of p28. These cytokines are of particular interest due to their potential to activate STAT1 and their relevance in the context of our study.

To address the question of what drives the enhanced STAT1 signaling, we are planning to perform ChIP experiments using these cytokines directly. This approach will allow us to evaluate their impact on STAT1 signaling more precisely, without relying on TCR + co-stimulation. By treating cells ex vivo with IL-35 and CLCF1, we aim to determine whether these cytokines are responsible for the increased STAT1 activity observed in Cd11c-p28-flox mice.

We acknowledge that ruling out autocrine IFN-γ production in vitro, as we have done, does not fully address the potential role of IFN-γ in vivo. Therefore, we are also considering additional in vivo experiments to further investigate this possibility. These studies will help us determine whether other sources of IFN-γ or other cytokines contribute to the observed STAT1 hyperactivation. Unfortunately, due to unforeseen challenges with mouse crosses, we anticipate a delay of approximately six months to obtain the appropriate genotypes necessary for these experiments. We are actively working to resolve these challenges and will update the manuscript with the results of these experiments upon completion.

(6) The RNAseq data affirms that SP, RTE, and Tn cells from Cd11c-p28-flox mice exhibit more STAT1 signaling than WT controls. However, this does little to explain the attendant hyper-Th1 phenotype. Is there evidence that epigenetic machinery is deregulated (to account for changes in DNA. histone methylation)? Were IFN-g and Tbet among these few observed DEG? If so, then this should be highlighted. If not, then the authors must address why not. Are there clues as to why STAT1 signing is exaggerated? Also, the hyper-STAT1 effect should be better described using more rigorous STAT1- and interferon-signature genesets (see the work of Virginia Pascual, Anne O'Garra).

Thank you for your valuable feedback and suggestions. We appreciate your interest in understanding the mechanisms underlying the hyper-Th1 phenotype observed in Cd11c-p28-flox mice. Below, we address each of your points in detail:

(1) Epigenetic Regulation:

We have conducted a thorough analysis of the global levels of key histone modifications, including H3K4me3, H3K9me3, and H3K27me3, as well as the mRNA expression of the enzymes responsible for catalyzing these marks. Our results indicate that there are no significant differences in these histone modifications or the expression of the associated enzymes between Cd11c-*p28f/f* and wildtype mice (Figure 3-figure supplement 1A-C). This suggests that the enhanced STAT1 signaling is not a consequence of broad epigenetic deregulation. Instead, we hypothesize that the observed changes may be driven by more specific molecular mechanisms, such as cytokine signaling pathways.

(2) IFN-γ and Tbx21 Expression:

Regarding the expression of Th1-associated genes, our analysis revealed a modest induction of *ifng* and *tbx21* (encoding T-bet) in the CD4SP population following TCR stimulation. However, the baseline expression levels of these genes were quite low in freshly isolated CD4SP cells. Specifically, *ifng* was undetectable, and *tbx21* had an FPKM of 0.29 in wildtype mice compared to 1.05 in *Cd11c-p28f/f* mice. While these findings indicate some upregulation of Th1-associated genes, the overall expression levels remain relatively low, suggesting that additional factors may contribute to the hyper-Th1 phenotype.

(3) STAT1 Signature Genesets:

We have revised our analysis to incorporate more rigorous STAT1 and interferon-signature genesets, as suggested. We have adopted gene sets from well-established studies, including those by Virginia Pascual and Anne O'Garra, to provide a more comprehensive and accurate assessment of STAT1 signaling. This approach has enhanced our ability to identify and characterize the genes involved in the STAT1 pathway, providing clearer insights into the exaggerated STAT1 signaling observed in our model.

We appreciate your guidance and are committed to refining our analysis to provide a more detailed understanding of the mechanisms driving the hyper-Th1 phenotype in Cd11c-p28-flox mice. We will continue to explore the potential roles of cytokines such as IL-35 and CLCF1, as well as other factors that may contribute to the observed changes in STAT1 signaling and Th1 differentiation. We look forward to sharing our updated findings and further discussing these mechanisms in our revised manuscript.

(7) Is the hyper-Th1 phenotype of SP, RTE, and Tn cells from Cd11c-p28-flox mice unique to the CD4 compartment? Are developing CD8^+^ cells similarly prone to increased STAT1 signaling and IFN-g production?

Thank you for raising this important point. Our data indeed suggests that the hyper-Th1 phenotype observed in SP, RTE, and Tn cells from *Cd11c-p28f/f* mice is unique to the CD4^+^ T cell compartment. Specifically, we found that while CD4^+^ SP cells from *Cd11c-p28f/f* mice exhibited a significant upregulation in IL-27 receptor expression (both IL27Ra and gp130) compared to wild-type (WT) mice, CD8^+^ SP cells from the same genotype showed markedly lower expression of these receptor subunits (Figure 1C in Sci Rep. 2016 Jul 29:6:30448. DOI: 10.1038/srep30448). This finding is further supported by our observation that the phosphorylation levels of STAT1, STAT3, and STAT4, downstream targets of IL-27 signaling, were comparable between CD8 SP cells from *Cd11c-p28f/f* and WT mice (Author response image 1). Additionally, we observed no significant difference in IFN-γ and granzyme B production between naïve CD8 T cells isolated from the lymph nodes of the two genotypes (Author response image 1). Taken together, these results suggest that the enhanced Th1 differentiation and IFN-γ production seen in the CD4^+^ T cell population from *Cd11c-p28f/f* mice is not recapitulated in the CD8^+^ T cell lineage.

**Author response image 2. sa3fig2:** (**A**) Intracellular staining was performed with freshly isolated thymocytes from *Cd11c-p28f/f* mice and WT littermates mice using antibodies against phosphorylated STAT1 (Y701), STAT3 (Y705), and STAT4 (Y693). The mean fluorescence intensity (MFI) for CD8 SP from three independent experiments (mean ± SD, n=3). (**B**) CD8^+^ naive T cells were cultured under Th0 conditions for 3 days. The frequency of IFN-γ-, and granzyme B-producing CD8^+^ T cells were determined analyzed by intracellular staining. Representative dot plots (left) and quantification (right, mean ± SD, n=6).

Minor points and questions(1) Line 84 - Villarino et al. and Pflanz et al. are mis-referenced. Neither involves Trypanosome studies. The former is on Toxoplasma infection and, thus, should be properly referenced in the following sentence.

Thank you for pointing out this error. You are correct that the references to Villarino et al. and Pflanz et al. were misapplied in the context of Trypanosome studies. Villarino et al. focuses on Toxoplasma infection, and we appreciate your guidance to ensure accurate citation. We will correct this in the manuscript and properly cite the studies in their appropriate contexts. Thank you for your vigilance in maintaining the accuracy of our references.

(2) T－bet protein should also be measured by cytometry

We sincerely thank the reviewer for the valuable suggestion regarding the measurement of T-bet protein levels. In response to this comment, we have performed additional experiments to quantify T-bet protein expression using flow cytometry. The results of these analyses have been incorporated into the revised manuscript as Figure 1F.

**Reviewer #2 (Recommendations For The Authors):**
(1) When new mouse strains are generated in this study, there is no comment on whether there are any changes in the frequency or cell number of CD4 T cells. For instance, in Aire-deficient CD11c-p28 floxed mice, it should be noted whether CD4SP, naïve CD4, and CD4 RTE are all the same in frequency and number compared to their littermate controls. Also, is there any effect on the generation of these thymocytes?

We sincerely thank the reviewer for raising this important point regarding the potential changes in the frequency and cell numbers of CD4^+^ T cells in the newly generated mouse strains. In response to the reviewer’s question, we would like to clarify the following:

(1) Impact of *Aire* deficiency on CD4^+^ T Cells:

As previously reported by us and others (Aging Dis. 2019, doi: 10.14336/AD.2018.0608; Science. 2002, doi: 10.1126/science.1075958), *Aire* deficiency does not significantly alter the overall number or frequency of CD4 single-positive (CD4SP) thymocytes, recent thymic emigrants (RTEs), or naïve CD4^+^ T cells. However, it profoundly affects their composition and functional properties, leading to the escape of autoreactive T cells and subsequent autoimmune manifestations.

(2) Observations in *Cd11c-p28f/fAire-/-* mice:

In our study, we observed that the number and frequency of CD4^+^ T cells in the spleen and lymph nodes were comparable among *Cd11c-p28f/f*, *Aire-/-*, and *Cd11c-p28f/fAire-/-* mice, and WT controls. This suggests that the genetic modifications did not significantly impact the overall development or peripheral maintenance of CD4^+^ T cells.

**Author response image 3. sa3fig3:** 

(3) Challenges in assessing RTEs in double knockout mice:

To accurately assess RTEs in the double knockout mice, it would be necessary to cross these mice with Rag-GFP reporter mice, which specifically label RTEs. However, breeding the appropriate mouse strain for this analysis would require additional time and resources, which were beyond the scope of the current study.

(2) There are a couple of typos throughout the manuscript. For example, line 91: IL-27Rα or line 313: phenotype.

We apologize for the typographical errors. We have carefully reviewed the entire manuscript and corrected all identified mistakes, including those on line 91 (IL-27Rα) and line 305 (phenotype).

(4) The authors should show each data point on their bar graphs.

Thank you for the suggestion. We have presented each data point on their bar graphs in the revised manuscript.

(4) It should be noted from which organs the RTE and the naïve T cells were harvested.

Thank you for the constructive suggestion. We isolated CD4^+^ RTEs and mature naive CD4^+^ T cells by sorting GFP^+^CD4^+^CD8^-^CD^-^NK1.1^-^ cells (RTEs) and GFP^-^CD4^+^CD8^-^CD^-^CD44^lo^ cells (naive T cells) from lymph nodes. This detail has been added to the manuscript on line 475.